



# The MIPAS global climatology of $BrONO_2$ 2002–2012: a test for stratospheric bromine chemistry

Michael Höpfner[1], Oliver Kirner[2], Gerald Wetzel[1], Björn-Martin Sinnhuber[1], Florian Haenel[1], Sören Johansson[1], Johannes Orphal[1], Roland Ruhnke[1], Gabriele Stiller[1], and Thomas von Clarmann[1]

[1]Karlsruhe Institute of Technology, Institute of Meteorology and Climate Research, Karlsruhe, Germany
[2]Karlsruhe Institute of Technology, Steinbuch Centre for Computing, Karlsruhe, Germany

**Correspondence:** M. Höpfner (michael.hoepfner@kit.edu)

**Abstract.** We present the first observational dataset of vertically resolved global stratospheric $BrONO_2$ distributions from July 2002 until April 2012, and compare them to results of the atmospheric chemical climate model EMAC. The retrieved distributions are based on space-borne measurements of infrared limb-emission spectra recorded by the Michelson Interferometer for Passive Atmospheric Sounding (MIPAS) on Envisat. The derived vertical profiles of $BrONO_2$ volume mixing ratios represent $10°$ latitude bins and three-day means, separated into sunlit and observations in the dark. The estimated uncertainties are around 1–4 pptv caused by spectral noise for single profiles as well as for further parameter and systematic errors which may not improve by averaging. Vertical resolutions range from 3 to 8 km between 15 and 35 km altitude.

All leading modes of spatial and temporal variability of stratospheric $BrONO_2$ in the observations are well replicated by the model simulations: the large diurnal variability, the low values during polar winter as well as the maximum values at mid- and high latitudes during summer. Three major differences between observations and model results are observed: (1) a model underestimation of enhanced $BrONO_2$ in the polar winter stratosphere above about 30 km of up to 15 pptv, (2) up to 8 pptv higher modelled values than observed globally in the lower stratosphere up to 25 km most obvious during night, and (3) up to 5 pptv lower modelled concentrations at tropical latitudes between 27 and 32 km during sunlit conditions. (1) is explained by the model missing enhanced $NO_x$ produced in the mesosphere and lower thermosphere subsiding at high latitudes in winter. This is the first time that observational evidence for enhancement of $BrONO_2$ caused by mesospheric $NO_x$ production is reported. The other major inconsistencies (2,3) between EMAC model results and observations are studied by sensitivity runs with a 1d model. These tentatively hint to a model underestimation of heterogeneous loss of $BrONO_2$ in the lower stratosphere, a too low simulated production of $BrONO_2$ during day as well as strongly underestimated $BrONO_2$ volume mixing ratios when loss via reaction with $O(^3P)$ is considered additionally to photolysis. However, considering the uncertainty ranges of model parameters and of measurements, an unambiguous identification of the causes for the differences remains difficult.

The observations have also been used to derive the total stratospheric bromine content relative to years of stratospheric entry between 1997 and 2007. With an average value of $21.2\pm1.4$ pptv of $Br_y$ at mid-latitudes where the modelled adjustment from $BrONO_2$ to $Br_y$ is lowest, the MIPAS data agree with estimates of $Br_y$ derived from observations of BrO as well as from MIPAS-Balloon measurements of $BrONO_2$.





# 1 Introduction

Besides chlorine, bromine is the major halogen constituent with anthropogenic and natural sources affecting stratospheric ozone (e.g. Sinnhuber et al., 2009; Engel et al., 2018). After Wofsy et al. (1975) had described the possible relevance of bromine for ozone, the important role of bromine nitrate ($BrONO_2$) within stratospheric bromine chemistry was proposed by Spencer and Rowland (1978). They noticed the much faster photolysis of $BrONO_2$ compared to $ClONO_2$, which is an important prerequisite for the effectiveness of bromine ozone destruction cycles compared to those of chlorine (Lary, 1997; Klobas et al., 2020).

$BrONO_2$ ist produced via the termolecular reaction (Burkholder et al., 2019, and references therein):

$$BrO + NO_2 \xrightarrow{M} BrONO_2 \tag{R1}$$

Due to its relatively short lifetime, the $BrONO_2$ concentration is coupled strongly to changes of $NO_2$ (Lary, 1996). The 1-$\sigma$ uncertainty factor of the reaction rate R1 as provided by Burkholder et al. (2015, 2019) is 1.2 at 298 K increasing to ~1.9 at stratospheric temperatures of 220 K.

The main loss process of $BrONO_2$ during day is photolysis (Burkholder et al., 2019, and references therein):

$$BrONO_2 + h\nu \rightarrow \text{Products} \tag{R2}$$

in which the products are $Br + NO_3$ and $BrO + NO_2$. The recommended quantum yields at wavelengths above 300 nm, being most important in the lower stratosphere, are 0.85 and 0.15, respectively. While in Sander et al. (2011) a combined uncertainty in cross sections and quantum yields of 1.4 is provided, the most recent evaluations (Burkholder et al., 2015, 2019) assign one wavelength-independent uncertainty factor of 1.2 (2-$\sigma$) to the cross sections.

Further loss of $BrONO_2$ is due to atomic oxygen (Soller et al., 2001):

$$BrONO_2 + O(^3P) \rightarrow BrO + NO_3 \tag{R3}$$

which occurs, like R2, only during day due to the necessary presence of $O(^3P)$. The 1-$\sigma$ uncertainty factor for the reaction coefficient varies between 1.25 at room temperature and 1.3 at 220 K (Burkholder et al., 2019). However, independent confirmation of the reaction parameters of R3 is pending (Burkholder et al., 2019).

At last, heterogeneous reactions can affect $BrONO_2$ concentrations, like hydrolysis in sulfuric acid aerosols (Burkholder et al., 2019, and references therein):

$$BrONO_2 + H_2O(s, l, H_2SO_4 \cdot nH_2O) \rightarrow HOBr + HNO_3 \tag{R4}$$

or in combination with halogens at surfaces, like:

$$BrONO_2 + HCl(H_2O(s), H_2SO_4 \cdot nH_2O) \rightarrow BrCl + HNO_3 \tag{R5}$$

where typical uncertainty factors of the gas/surface reaction probabilities are in the range of 2–4 (Burkholder et al., 2019).





Given the relatively large uncertainties in most of these leading reactions involving $BrONO_2$, comparison of observations to model calculations can be helpful for verification or even for suggesting improvements. For example, Kreycy et al. (2013) analysed stratospheric balloon observations and concluded that the ratio $J_{BrONO_2}/k_{BrO+NO_2}$ should be increased to fit their data. Such investigations can be useful, first, to improve model simulations of stratospheric ozone loss and, second, to aid the analysis of the total stratospheric bromine ($Br_y$) content from observations of one species, such as BrO (e.g. Harder et al.,

2000; Dorf et al., 2006a, 2008; Millán et al., 2012; Stachnik et al., 2013; Kreycy et al., 2013; Werner et al., 2017; Engel et al., 2018).

    Anthropogenic and natural emissions both contribute roughly equally to the present-day stratospheric bromine loading: Engel et al. (2018) give a best estimate of the total stratospheric bromine loading for 2016 of 19.6 pptv, of which natural sources contributing slightly more than 10 pptv. Very short-lived source gases, such as bromoform ($CHBr_3$) and dibromomethane

($CH_2Br_2$) contribute about 5 pptv to the stratospheric bromine loading, but their precise current contribution, any possible long term changes and the additional influx of inorganic product gases (product gas injection, PGI) are still uncertain (e.g. Sinnhuber et al., 2009; Aschmann and Sinnhuber, 2013; Falk et al., 2017). In this context, the observation of $BrONO_2$ provides an additional independent approach to determine total $Br_y$ (Wetzel et al., 2017) and in consequence to estimate the relative contribution of very short-lived bromine substances (VSLS).

Due to its spectral lines in the microwave and UV-vis, remote sensing observations of BrO, the major inorganic bromine species in the lower stratosphere during sunlit hours, are common from ground (e.g. Solomon et al., 1989; Carroll et al., 1989; Fish et al., 1995; Theys et al., 2007; Hendrick et al., 2008), from aircraft (e.g. Koenig et al., 2017; Werner et al., 2017), from balloons (e.g. Harder et al., 2000; Pundt, 2002; Dorf et al., 2006a, 2008; Stachnik et al., 2013; Kreycy et al., 2013) and from satellites (e.g. Sinnhuber et al., 2005; Livesey et al., 2006; Kovalenko et al., 2007; McLinden et al., 2010; Rozanov et al.,

2011; Millán et al., 2012; Parrella et al., 2013). In contrast, $BrONO_2$, the most important night-time reservoir of bromine, was detected in infrared limb-emission observations by the Michelson Interferometer for Passive Atmospheric Sounding (MIPAS) instrument on board the Envisat satellite only a decade ago (Höpfner et al., 2009). At that time the retrieval of altitude profiles was complicated by uncertainties in the infrared absorption cross-sections of $BrONO_2$. In the meantime, Wagner and Birk (2016) provided an improved infrared spectroscopic database covering stratospheric conditions. On basis of these new data,

Wetzel et al. (2017) analysed the diurnal variation of $BrONO_2$ during three flights of the MIPAS-Balloon instrument.

    In this paper we introduce the first day- and night-time climatology of stratospheric $BrONO_2$ as derived for the ~10 years lifetime of MIPAS/Envisat. We compare the results to global model simulations and discuss major differences by use of 1d photochemical modelling. Finally, the total stratospheric $Br_y$ content is estimated.

## 2    Methods

### 2.1    MIPAS instrument and data analysis


Flying on the polar orbiting satellite Envisat, the limb-sounder MIPAS recorded infrared spectra of the atmospheric thermal emission from 2002 until 2012 (Fischer et al., 2008). MIPAS was operated in two major modes: during period 1 (P1), be-





tween July 2002 and March 2004, the spectral resolution, defined here as $0.5\times$ (maximum optical path difference)$^{-1}$, was $0.025\,\mathrm{cm}^{-1}$, and during period 2 (P2) between January 2005 and April 2012, the resolution was set to $0.0625\,\mathrm{cm}^{-1}$. During

P1 the spectra of the "nominal" viewing modes as used in this work were taken at 17 tangent points between 7 and 72 km with 3 km steps up to 42 km, and somewhat larger steps above. During P2, 27 spectra were recorded per limb-scan with latitude-dependent tangent altitudes ranging from 5–70 km at the poles to 12–77 km over the Equator, with steps increasing with height from 1.5 km to 4.5 km. The along-track sampling distance between each limb-scan was ∼550 km during P1 and ∼420 km during P2. The local solar equator crossing time at the position of the tangent points is around 10:10 for the descending node

and 22:20 for the ascending node of the sun-synchronous orbit.

     Since the first stratospheric detection of $BrONO_2$ (Höpfner et al., 2009), retrieval from averaged MIPAS spectra has been established for species with very weak signatures, such as $SO_2$ and $NH_3$ (Höpfner et al., 2013, 2016). The retrieval of vertical profiles of $BrONO_2$ volume mixing ratios as applied for the current dataset follows closely the procedure described in Höpfner et al. (2009). Here we shortly describe the retrieval scheme as well as the applied improvements with respect to Höpfner et al.

100    (2009):

     For the selection of spectra to be averaged zonally as well as temporally, the cloud filter method by Spang et al. (2004) has been applied to sort out any measurements affected by tropospheric as well as polar stratospheric clouds. Further, only spectra above about 15 km tangent altitude have been used for averaging so as to concentrate mainly on the stratosphere. We have applied a constrained nonlinear multiparameter least-squares fitting procedure to each limb-sequence of averaged spectra to

derive profiles of trace gas volume mixing ratios at 1 km spaced vertical levels. Here we have used the same spectral interval ($801$–$820\,\mathrm{cm}^{-1}$) and atmospheric parameters simultaneously fitted with $BrONO_2$ ($O_3$, $ClONO_2$, $NO_2$, $CFC{-}22$, $HNO_4$, $COF_2$, $HNO_3$, $ClO$ $CCl_4$, $CFC{-}113$, PAN, T) as in Höpfner et al. (2009). We have applied a first-order smoothing constraint (Steck, 2002; Tikhonov, 1963) to dampen oscillations in the retrieved profiles. The regularization strength for each of the simultaneously derived species has been adjusted separately and the related a-priori profile for the target species $BrONO_2$ was

set to zero while for the other species climatological profiles have been used.

     Major improvements and updates compared to Höpfner et al. (2009) are:

1. The most recent version (V8.03) of level-1B calibrated limb-radiances by ESA has been used (https://earth.esa.int/web/sppa/mission-performance/esa-missions/envisat/mipas/products-availability/level-1/level1-8.03).

2. To simulate the spectral feature of $BrONO_2$, the new pressure- and temperature-dependent infrared spectroscopic

115       database by Wagner and Birk (2016) have been used.

3. The spectroscopy of the interfering gases has been taken from the HITRAN2016 database (Gordon et al., 2017) with the exception of $HO_2NO_2$. For this gas, the infrared cross-sections in HITRAN for 220 K by May and Friedl (1993) have been extended by the ones of Friedl et al. (1994), which were measured at 298 K, to account for different atmospheric temperatures by two-point interpolation (Wetzel et al., 2017).



4.  While in Höpfner et al. (2009) retrievals have been performed based on monthly mean spectra of September 2002 and 2003 within few coarse latitude bands, here we have subdivided the MIPAS measurements into 18 latitudes bands of 10° spacing with a temporal binning of three days over the whole observational period 2002–2012.

We have estimated altitude dependent errors of the $BrONO_2$ retrieval by applying assumptions about single error sources to two randomly selected periods in March and June for the years 2003 and 2009, thus during P1 and P2, respectively. The results are shown in Fig. 1 together with the total error profile calculated by quadratic combination of single error components. Instrumental uncertainties are estimated at 3% for the instrument line shape expressed as linear loss of modulation efficiency toward the maximum optical path difference of the interferometer (ILS), 1% for radiometric gain calibration (RadGain), and 300 m for tangent height knowledge (Htang). The uncertainty of ECMWF temperatures (Temp) has been set to values of 2 K below and 5 K above 35 km altitude. The uncertainty of the $BrONO_2$ spectroscopy has been assumed as 5%, which is on the conservative side considering the 2% ($1-\sigma$) error estimation given in Wagner and Birk (2016). Further errors refer to the spectroscopic parameters of interfering gases. For those, we have assumed uncertainties of 5% for species described by cross-sections (SpecXitf), 5% for intensities (SpecINTitf) and 10% for the half widths of the line-parameters (SpecHWift). These assumptions are within the typical errors provided in the database (Gordon et al., 2017). A further error term describing the retrieval from averaged spectra (NonLin) has been estimated on the basis of dedicated retrieval simulations as detailed in Höpfner et al. (2009). For this estimate the values used for the tangent altitude scatter of single observations were set to 400 m and 300 m ($1-\sigma$) during P1 and P2, respectively.

The total error estimate, as calculated by quadratic combination of the single components is given by the blue lines in Fig. 1. Around that, the blue shading indicates the variability of the estimated errors between all latitude bands. In general, the estimated total errors vary between about 1 pptv and 4 pptv, independent of day- or night-time observations. They appear to be slightly smaller during P1 compared to P2, which is probably due to the better spectral resolution during P1.

As a further diagnostic measure of the retrieval, Fig. 2 shows the vertical resolution as a function of altitude. It has been calculated by dividing the retrieval grid width of 1 km by the diagonal elements of the averaging kernel matrices (Rodgers, 2004). The vertical resolution is about 3 km at 15 km altitude and becomes coarser with altitude, reaching 8 km at 35 km altitude. The vertical resolution is generally finer at the tangent altitudes and coarser at retrieval levels between the tangent altitudes. Conversely, the retrieval noise is larger at the tangent altitudes and smaller at altitude levels in between. These effects are only visible when a retrieval setup is chosen where the retrieval grid is finer than the tangent altitude spacing. Since in period P1 the tangent altitude grid is fixed, this effect survives the averaging, leading to a zigzag profile of the vertical resolution. In contrast, in P2 the tangent altitude grid varies with latitude, and the zigzag features of vertical resolution average out.





**Figure 1.** Retrieval error estimates from two 3-day periods during P1 (top) and two periods during P2 (bottom) for dark (left) and sunlit (right) conditions. Considered error sources are the uncertainties of the instrumental line shape and radiometric gain calibration (ILS, RadGain), the pointing knowledge (Htang), assumed temperature profiles (Temp), spectroscopic errors of $BrONO_2$ absorption cross sections (SpecBrONO2) and errors in cross-sections (SpecXitf), line half-widths (SpecHWift) and line intensities (SpecINTitf) of interfering species, as well as the error due to the applied technique of retrievals from averaged spectra (NonLin) and the spectral noise of the instrument (Noise). The total error (Total Err) has been determined by quadratic combination of all single error components while the combined parameter and systematic error (Tot paraerr) considers all uncertainties except the spectral noise. The blue filled space around the total error curves indicates the areas in which 90% of the total error profile estimates fall into.



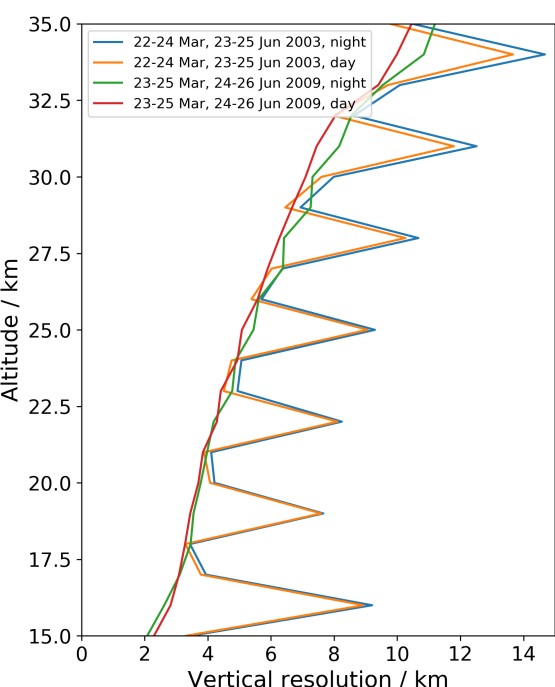

**Figure 2.** Examples for the vertical resolution of the MIPAS $BrONO_2$ retrieval as derived from the diagonal elements of the averaging kernel matrices. Given curves are averages over all latitude bands for the given periods during P1 and P2. The zigzag during P1 is caused by the constant tangent altitude grid while during P2, the variation of tangent altitudes with latitude smear out this effect. Note that the vertical resolution is generally finest at the tangent points and coarsest in between.





## 2.2 Atmospheric modelling

We have compared the MIPAS $BrONO_2$ dataset with a multi-annual simulation from the chemical climate model ECHAM/
MESSy Atmospheric Chemistry (EMAC) (Jöckel et al., 2010). Within EMAC, the interface Modular Earth Submodel System
(MESSy) links the sub-models describing tropospheric and middle atmospheric processes to the dynamical core, the fifth-
generation European Centre Hamburg general circulation model ECHAM5 (Roeckner et al., 2006). We have used EMAC
(ECHAM5 version 5.3.02, MESSy version 2.52) in the T42L90MA-resolution with 90 vertical hybrid pressure levels from the
ground up to $0.01\,hPa$ ($\sim 80\,km$) and a horizontal resolution of $\sim 2.8° \times 2.8°$ latitude $\times$ longitude. The sub-models MECCA
(Sander et al., 2005) and MSBM (Kirner et al., 2011) simulate gas-phase chemistry and polar stratospheric clouds including
heterogeneous reaction rates, respectively. To reproduce realistic conditions for comparison with the observations, the model
run was nudged towards the ECMWF reanalysis ERA-Interim (Dee et al., 2011) by a Newtonian relaxation technique of surface
pressure, temperature, vorticity, and divergence above the boundary layer and below $1\,hPa$ (van Aalst, 2005). We have applied
a comprehensive chemistry set-up from the troposphere to the lower mesosphere with more than 100 species involved in gas-
phase, photolysis, and heterogeneous reactions on liquid sulfate aerosols, nitric acid trihydrate (NAT), and ice particles. Rate
constants for gas phase reactions have been taken mainly from Atkinson et al. (2007) and the Jet Propulsion Laboratory (JPL)
compilation (Sander et al., 2011). Photochemical reactions of short-lived bromine containing organic compounds $CHBr_3$,
$CH_2Br_2$, $CH_2ClBr$, $CHClBr_2$, and $CHCl_2Br$ are included in the model set-up (Jöckel et al., 2016). Boundary conditions for
$CH_3Br$ and the bromine-containing halons were taken from Meinshausen et al. (2011) and extended with the RCP6.0 scenario
as suggested by Eyring et al. (2013). We have used scenario 5 of Warwick et al. (2006) to describe the surface emissions of
these organic bromine species. During the MIPAS measurement periods, from the model output first all data within one hour
around 10 LT and 22 LT were selected. Depending on their latitude, longitude and altitude, they were then assigned to day-
and night-time conditions and averaged over the observational bins of 10°latitude and three-day periods.

For specific sensitivity investigations at low latitudes we have applied a 1d photochemical stacked box model. The chemical
mechanism of the 1d model is based on the SLIMCAT model (Sinnhuber et al., 2005, and references therein). The 1d model
runs have been initialized with equatorial mean profiles of the EMAC simulation of all inorganic $Br_y$ species and $NO_2$. For
the other species as well as pressure and temperature, equatorial profiles from the MIPAS reference database have been used
(Remedios et al., 2007). A comparison of the parameters of several bromine reactions between EMAC, the 1d baseline model
run and the JPL2019 compilation (Burkholder et al., 2019) is provided in Tab. D1.





## 3 The MIPAS dataset in comparison to EMAC model results

### 3.1 Overview of the measurements

We provide overviews of the MIPAS $BrONO_2$ volume mixing ratio datasets in full temporal resolution in the left panels of Figs. 3 and 4 for observations during dark (night) and sunlit (day) conditions, respectively. White spaces indicate regions where

no measurements are available, such as observations in the dark during high-latitude summer as well as sunlit measurements during winter. A measurement gap due to instrumental issues of MIPAS happened between April 2004 and January 2005 and in the following, observations were ramped up until about 2007. From then on, quasi continuous coverage exists until April 2012. The coverage at lower altitudes is determined by the lower limit of $15\,\mathrm{km}$ to confine the retrievals mainly on the stratospheric situation, by the cloud coverage and the scan pattern of MIPAS restricting the dataset mainly in the tropics. Some additional

data gaps exist at high southern latitudes during winter when thick polar stratospheric clouds (PSCs) obscured the observations.

From Figs. 3 and 4 the major features of the stratospheric $BrONO_2$ variability can be discerned in our measurements:

1. The diurnal variability (Fig. 3 vs. Fig. 4) as a manifestation of the fast photolysis during day (Eq. R2) versus the production (Eq. R1).

2. The annual recurrence of low values during night at high latitudes (Fig. 3) due to the lack of $NO_x$ as supply for the
production (Eq. R1) in combination with heterogeneous loss at PSC particles (Eqs. R4 and R5).

3. The annual maxima of $BrONO_2$ volume mixing ratios at high- and mid-latitudes during day- and night-time observations in summer caused by the annual variability of $NO_2$.

4. The lack of a similarly outstanding annual signal at tropical latitudes.

We have reduced the dataset to annual views by averaging over the whole MIPAS observational period in order to provide a
clearer picture on intra-annual variabilities. The related horizontal and vertical cross-sections are presented in Figs. 5–8.

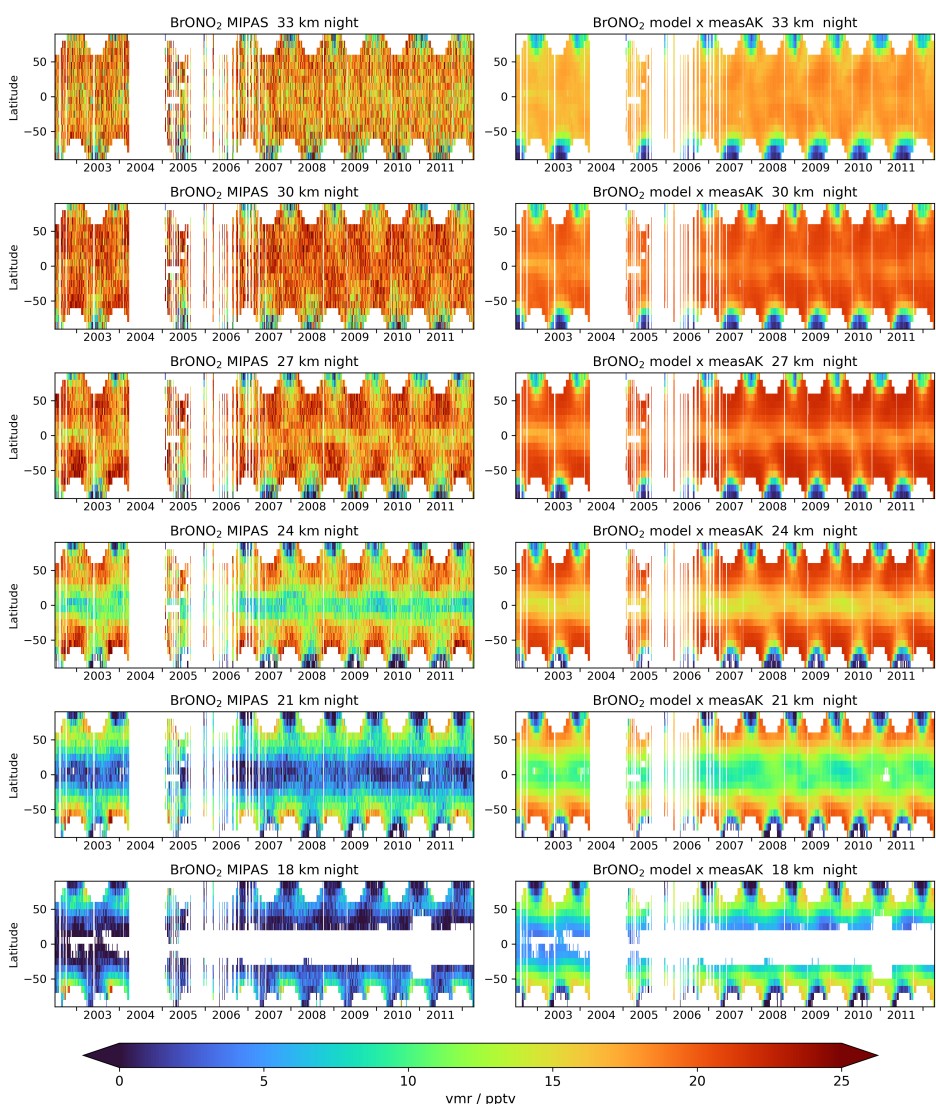

**Figure 3.** Horizontal cross sections (latitude versus time) of measured (left) and modelled (right) $BrONO_2$ volume mixing ratios at selected altitudes over the whole time period of MIPAS observations during dark conditions. Retrieval averaging kernels have been applied to model data. White areas indicate the absence of measurements.



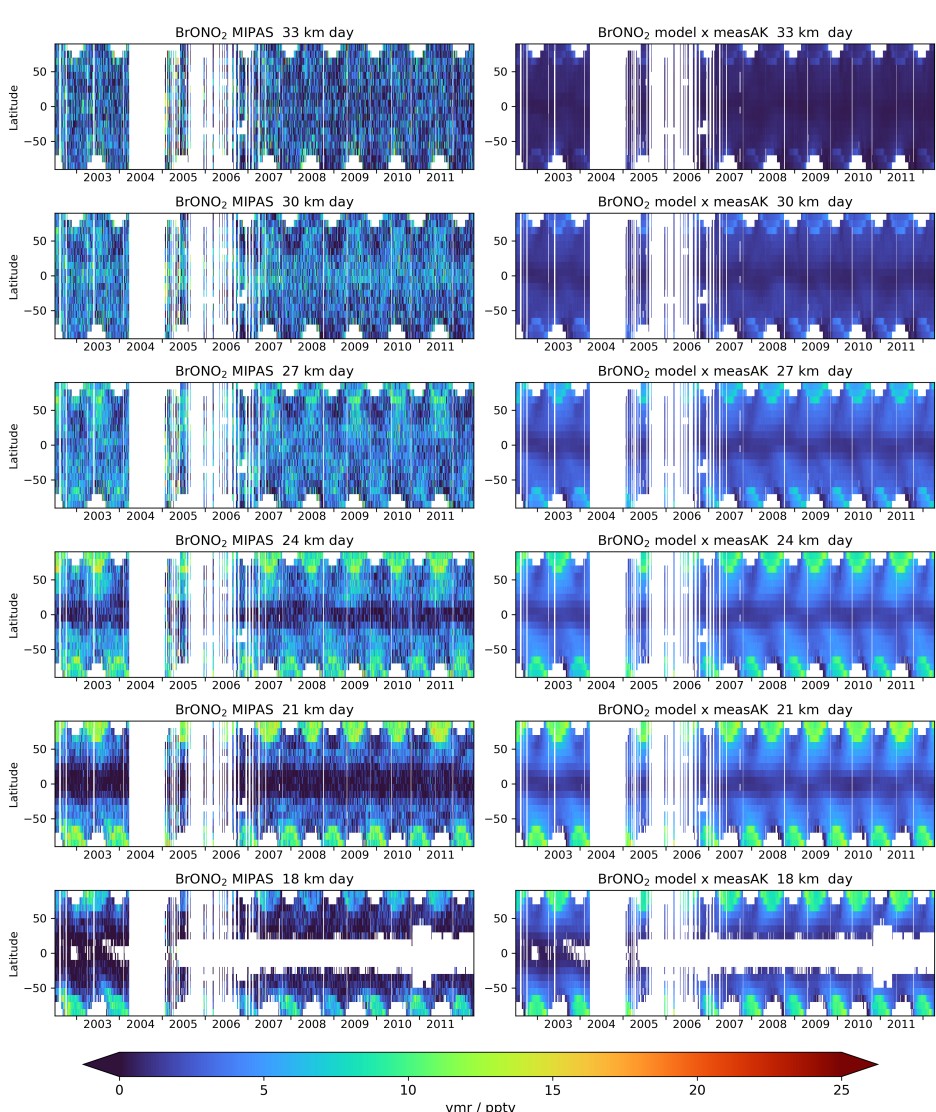

**Figure 4.** Same as in Fig. 3 but for sunlit measurements.



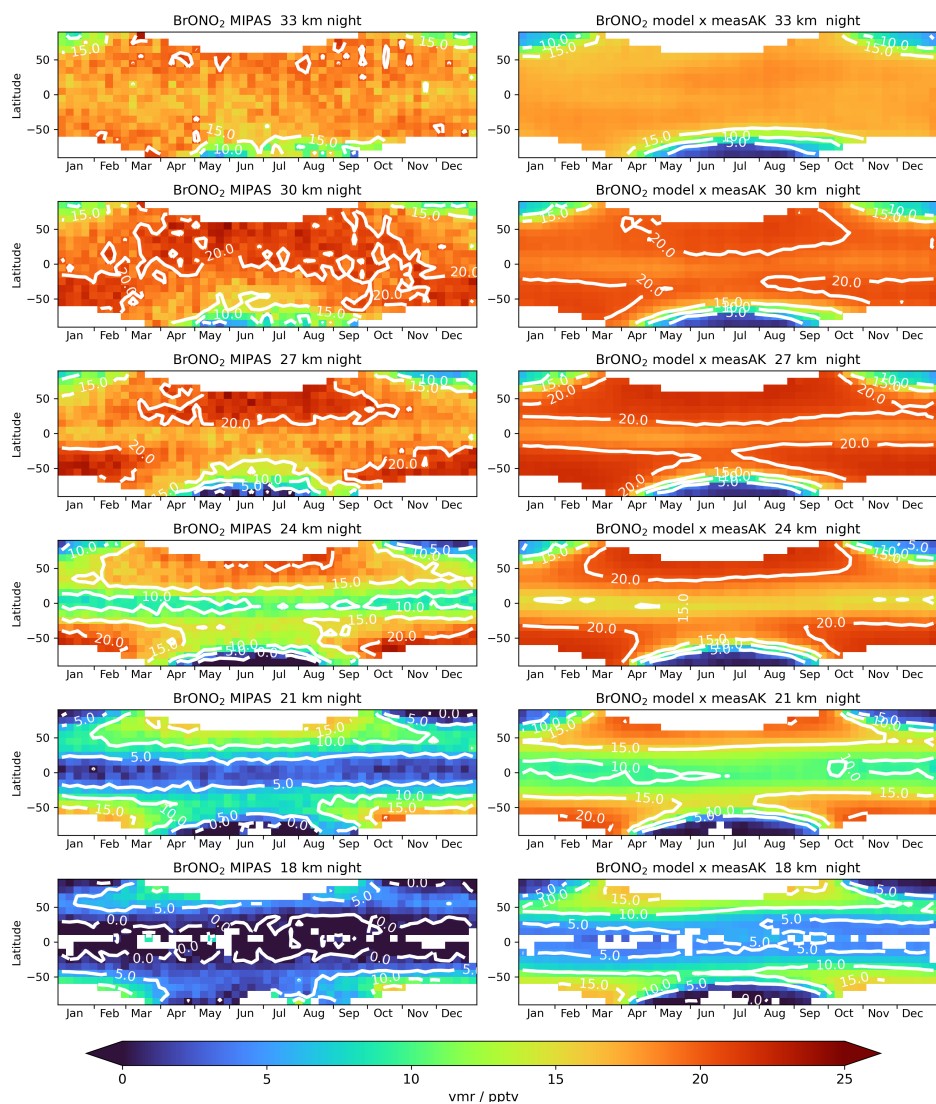

**Figure 5.** Horizontal cross-sections (latitude versus time) of the annual development of measured (left) and modelled (right, with averaging kernels applied) $BrONO_2$ volume mixing ratios during dark conditions at selected altitudes calculated as average values over the whole MIPAS dataset in weekly bins (see Fig. 3).



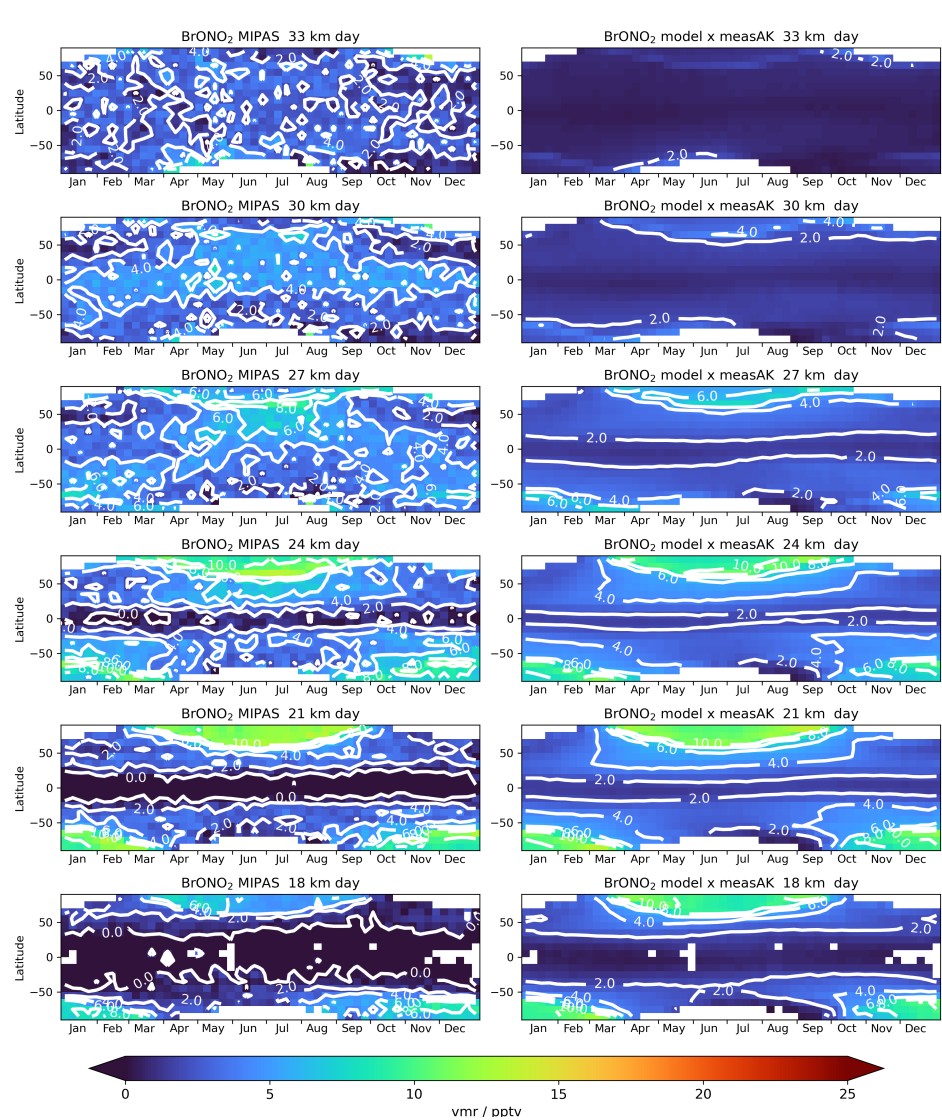

**Figure 6.** Same as Fig. 5 but for sunlit measurements (see Fig. 4).



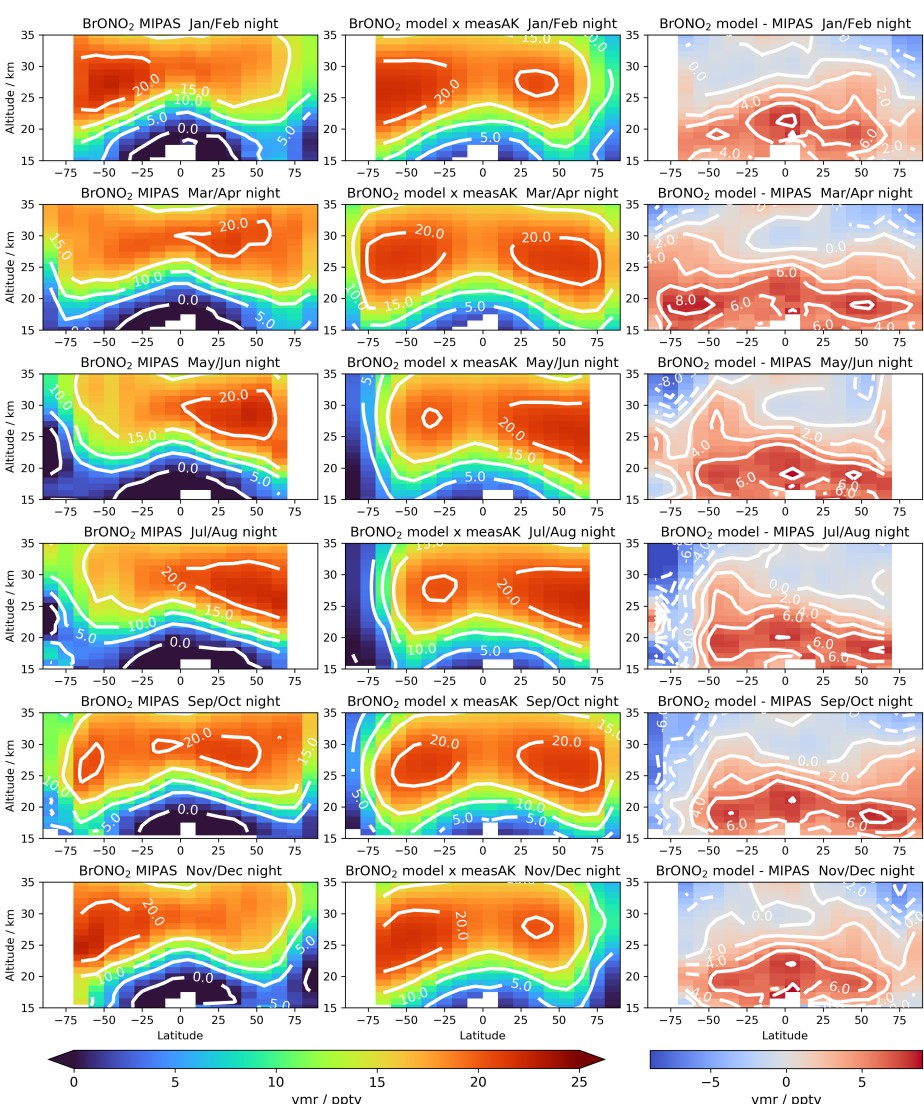

**Figure 7.** Bimonthly averaged cross-sections (altitude versus latitude) of $BrONO_2$ volume mixing ratios for dark conditions. Left: measurements, middle: model with retrieval averaging kernels applied, right: model minus MIPAS.



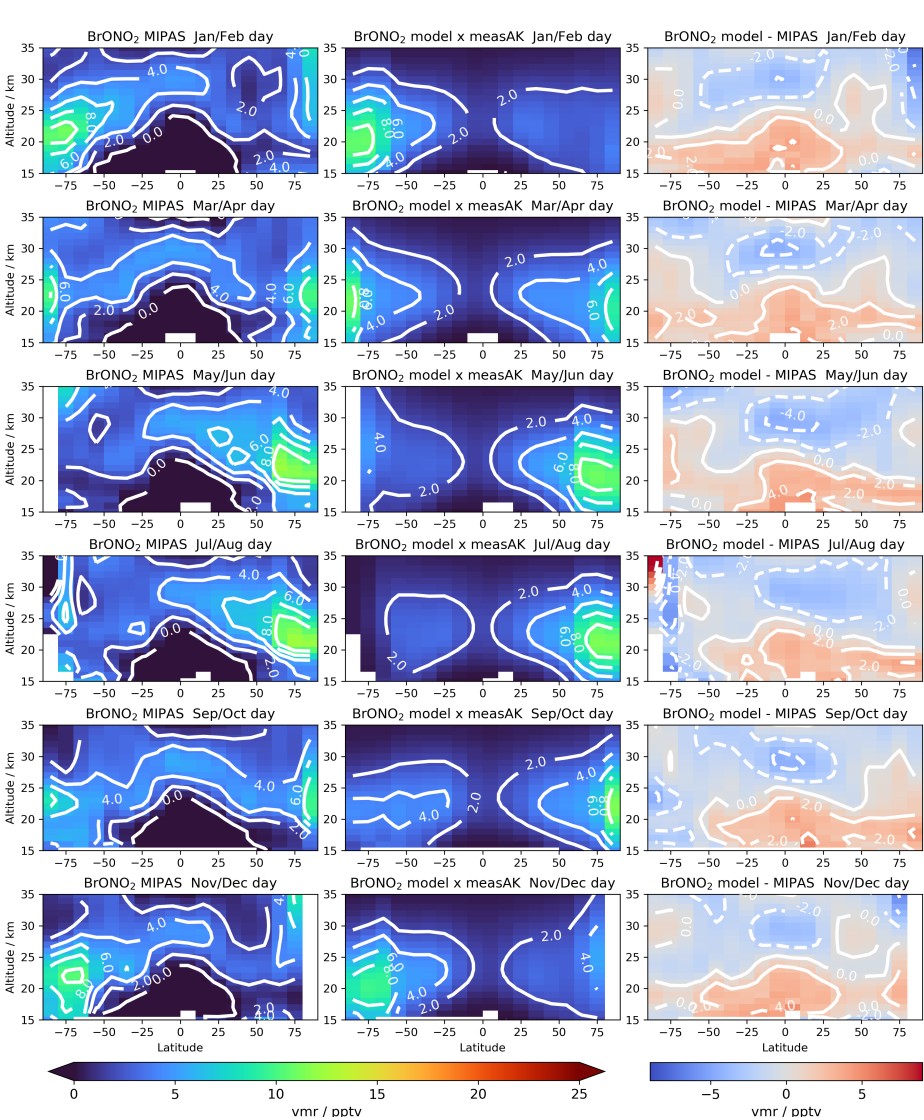

**Figure 8.** Same as Fig. 7 but for sunlit conditions.





## 3.2 Measurement/model comparisons

The results of the EMAC model run are presented in the second column in each of the Figures 3–8. For these comparisons, we have applied the averaging kernel matrix of each retrieval to the related modelled profile. Differences between model results with and without averaging kernel application are generally below 1 pptv with only sporadic exceptions at highest altitudes near polar latitudes where differences reach up to 3–4 pptv (Figs. A1 and A2).

From comparing measured and modelled distributions of $BrONO_2$ volume mixing ratios in Figs. 3–8 it is evident, that the model reproduces all major modes of variability which are present in the observations as described in the previous section. Despite this agreement, there are a few areas where systematic deviations are prominent.

The most obvious differences appear (1) at higher altitudes during polar winter, (2) in the lower stratosphere mainly at mid- and low latitudes during the entire year as well as (3) at altitudes around 30 km in the tropics during sunlit conditions. We discuss these differences one by one below.

### 3.2.1 Low modelled polar winter $BrONO_2$

One disagreement between modelled $BrONO_2$ and MIPAS observations can best be observed in Figs. 5 and 7: at altitudes of around 30 km and above for latitudes south of 70°S, the model predicts values of less than 5 pptv from May until September. The corresponding measurements, however, reach values of 10 to 15 pptv. While not as pronounced, this feature is also visible in the Arctic wintertime stratosphere with model estimates of 5–10 pptv and measurements of 10–15 pptv. We explain these low model values of $BrONO_2$ by an underestimation of $NO_2$ as visible in Fig. B1. In this Figure, the MIPAS $NO_2$ distributions are compared to the EMAC model results of $NO_2$. The missing $NO_2$ in the simulations is due to an insufficient supply of $NO_x$ through downward propagation from the upper mesosphere and lower thermosphere. This stratospheric enhancement of $NO_x$ through production by energetic particle precipitation and downwelling during polar winter has been investigated e.g. on basis of MIPAS observations by Funke et al. (2005, 2014).

### 3.2.2 High modelled night-time $BrONO_2$ at lower altitudes

As visible e.g. in Figures 5 and 7, the model overestimates $BrONO_2$ volume mixing ratios with respect to the MIPAS results by up to 8 pptv in the lower stratosphere at altitudes up to about 25 km during night. Such differences are also present during day, albeit to a smaller extent (up to about 4 pptv, see Fig. 8).

The black line in panel "night" of the top row in Fig. 9 represents the measured mean night-time profile over the whole period at 5°S. The related EMAC model result is shown in red. Maximum differences between both are about 8 pptv at around 20–22 km altitude. As possible explanation for these differences we will discuss below (1) measurement errors, (2) wrong modelling of the release of $Br_y$ from its source gases and, (3) wrong partitioning of $Br_y$ between its main constituents in the model simulations.

1. The degree of discrepancy between measurement and model of up to 8 pptv over an altitude range of around 5 km cannot reasonably be explained given the errors estimated for the MIPAS $BrONO_2$ retrieval (Fig. 1). This would allow





for discrepancies of around 2–3 pptv, especially considering that the "noise" error term is strongly reduced by the temporal averaging over all equatorial measurements. Still one cannot rule out unequivocally any unidentified additional systematic error source in the measurements contributing to these differences.

2. Dorf et al. (2008) report on balloon observations of BrO in the framework of an ENVISAT validation campaign in Teresina, Brasil (5.1°S, 42.9°W) on 17 June 2005. In the altitude region of 20–22 km they observed volume mixing ratios of BrO of around 6–10 pptv which agrees with our EMAC model results during daytime (see Fig. 9, 2nd row, red line in panel 'day') that indicates 8–11 pptv of BrO in the same altitude range. Similar mixing ratios of BrO at these altitudes in the tropics have also been reproduced by other model simulations (e.g., Theys et al., 2009) as well as by satellite observations (Sinnhuber et al., 2005; Sioris et al., 2006; Rozanov et al., 2011; Parrella et al., 2013). A possible contribution to a model overestimation might also be the used emission scenario of organic bromine species by Warwick et al. (2006). As has e.g. been shown by Keber et al. (2020), this scenario probably leads to an overestimation of brominated VSLS by up to 2 pptv, which is, however, not enough to explain the observed differences. Thus, it is highly unlikely that the inorganic bromine content at 20–25 km is strongly overestimated in the EMAC model calculations.

3. The modelled partitioning of $Br_y$ at those altitudes during night is essentially determined by the heterogeneous conversion of $BrONO_2$ into HOBr at sulfate aerosols (Eq. R4). For a more detailed investigation we have performed sensitivity simulation with the 1d model the results of which are shown in Fig. 9. The rate coefficient for heterogeneous reactions is proportional to the aerosol surface area density and the reaction probability $\gamma$. For the baseline 1d calculation, we have applied a vertical profile of aerosol surface area densities which was derived from the mean tropical aerosol volume densities as available from MIPAS (Günther et al., 2018) assuming a lognormal size distribution with a number density of $10\,cm^{-3}$ and a width of 1.8. Figure 10 provides a comparison of the resulting "MIPAS mean" surface area densities with SAGE II profiles in the tropics between July 2002 and August 2005 (Damadeo et al., 2013) as well as in-situ measurements at mid-latitudes over Laramie (41°S, 105°W) between 2002 and 2012 (Deshler et al., 2019). The reaction probabilities for the 1d baseline run (dashed pink curve in right panel of Fig. 10) were determined according to Hanson (2003) as reported by Burkholder et al. (2019).

A test without consideration of heterogeneous conversion of $BrONO_2$ into HOBr (dotted orange line in Fig. 9) results in a night-time increase of $BrONO_2$ by up to 5 pptv as well as the corresponding decrease of HOBr. To replicate the observations of $BrONO_2$ we performed two tests by adjusting the profile of aerosol surface area density: one (a) where we kept the altitude-dependent reaction probabilities (dashed pink curves in Figs. 9 and 10) and one (b) where the reaction probabilities were set to one over the entire altitude range (dashed-dotted purple curves in Figs. 9 and 10). As can be seen from Fig. 10, in test (a) one would require surface area densities enhanced by factors of more than 3 to deplete $BrONO_2$ to such an extent that the observations in the altitude range 20–26 km are met. For the idealized case of $\gamma=1$, the aerosol surface area densities (SAD) had to be adjusted up to 24 km only, however, still by a factor of more than 3 at 20 km altitude.



In summary, we have to conclude that there is no compelling evidence for any of the three explanations above causing the observed differences between measurements and model results. While (3) would imply an increase of aerosol surface area density contradicting related observations, (2) would oppose observations of BrO, and (1) seems out of reach within our estimated retrieval errors for $BrONO_2$ from MIPAS.

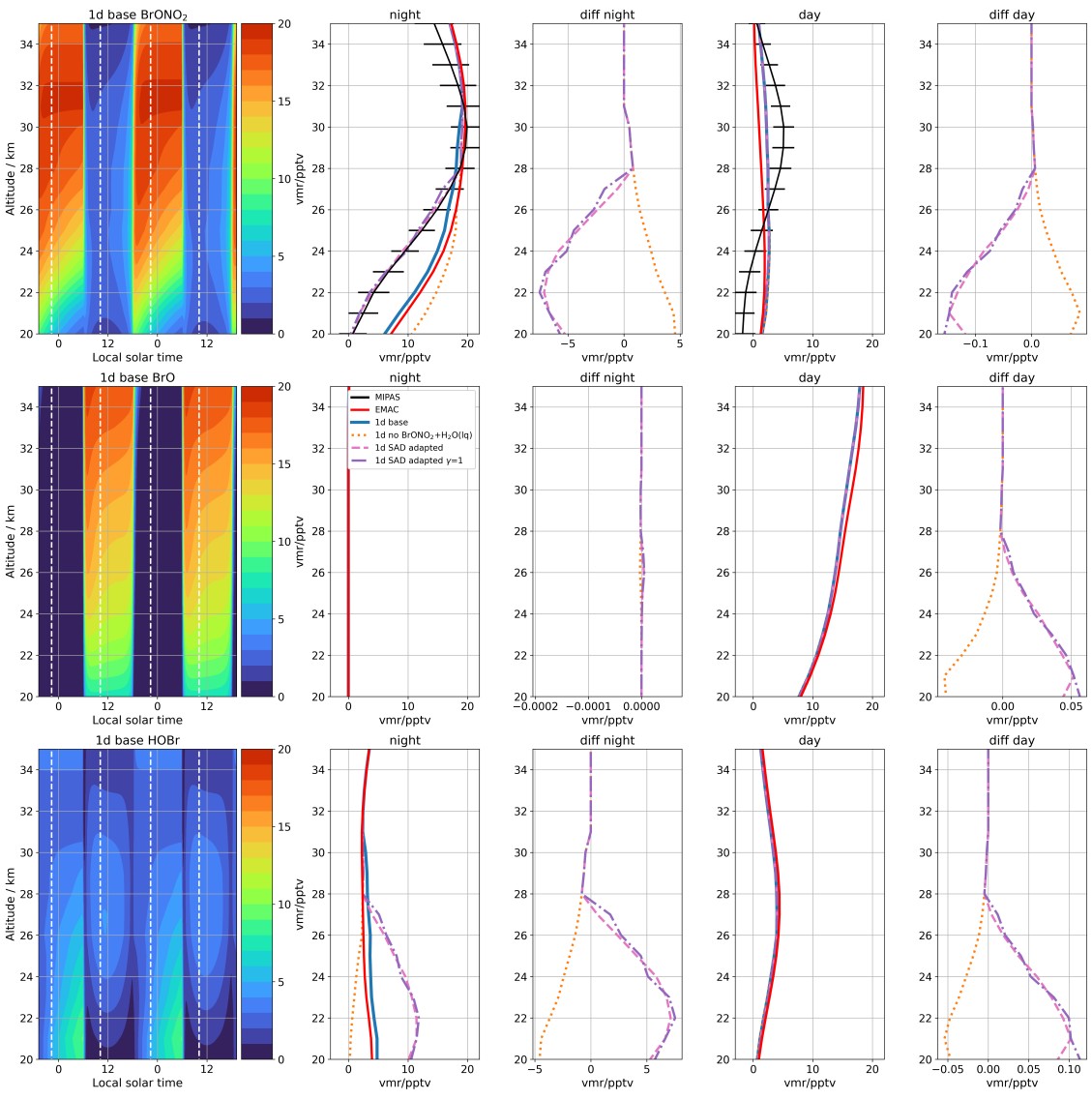

**Figure 9.** Sensitivity 1d model simulations for testing of the night-time EMAC model overestimation of $BrONO_2$ below 27 km altitude for $BrONO_2$ (top row), $BrO$ (middle row) and $HOBr$ (bottom row). Left: diurnal evolution of the "1d base" run for a period of two days. The white dashed lines indicate the local solar times of the MIPAS observations. The columns "night" and "day" contain the averaged night-time and daytime profiles of MIPAS at 5°S as black solid lines along with error bars indicating the 2-$\sigma$ estimated measurement uncertainty. The mean EMAC results are provided in red. The other curves illustrate the results of the 1d model simulations and the columns "diff night" and "diff day" show the differences between the 1d sensitivity runs and the "1d base" run.



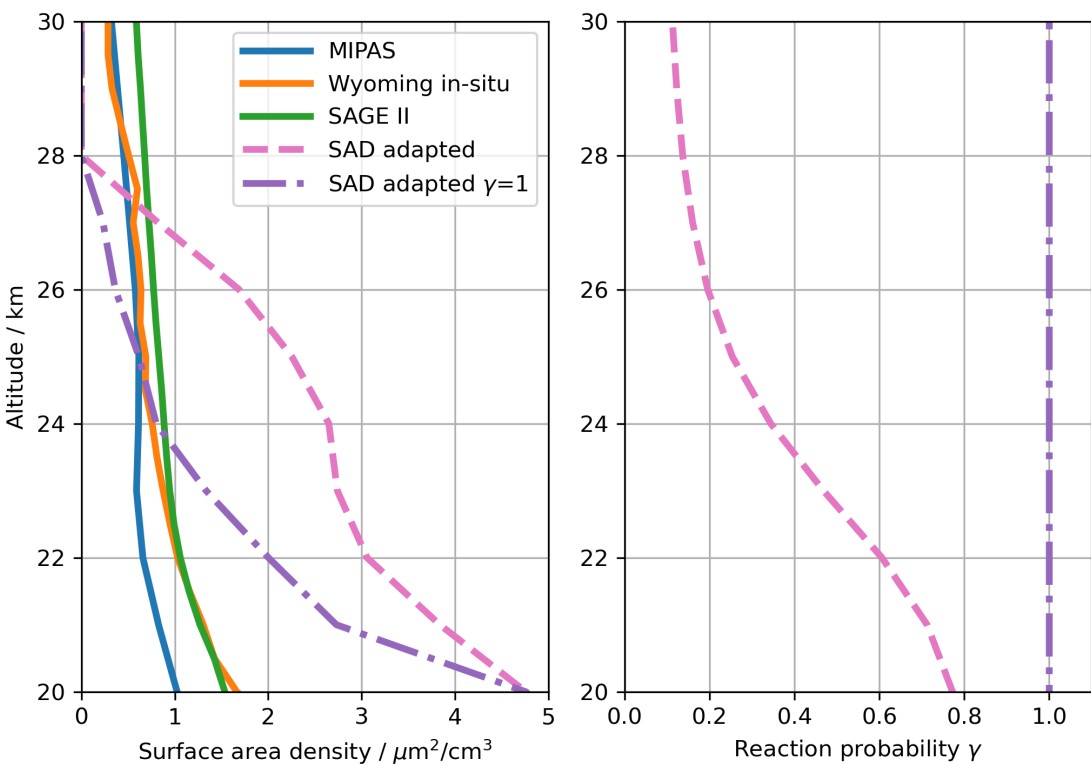

**Figure 10.** Left: vertical profiles of surface area densities (SAD). Blue: mean MIPAS tropical profile (Günther et al., 2018) (see text for details), orange: mean profile from in-situ observations over Laramie (41°S, 105°W) between 2002 and 2012, green: mean SAGE II profiles in the tropics between 2002 and 2005 (Damadeo et al., 2013). The dashed pink and dash-dotted purple lines show the adapted SAD profiles corresponding to the 1d model results indicated by the same line styles in Fig. 9. Right: reaction probability profiles for hydrolysis of $BrONO_2$ (Eq. R4). The dashed pink profile refers to the standard one determined in the 1d model (Hanson, 2003; Burkholder et al., 2019) and the dash-dotted purple curve has been used in the sensitivity analysis to adjust SAD (left panel and Fig. 9).





### 3.2.3 Low modelled daytime $BrONO_2$ at low latitudes

The EMAC model calculations fit quite well the observed night-time $BrONO_2$ maximum at 29–30 km altitude in the tropics (see Fig. 7). However, during daytime (Fig. 8), $BrONO_2$ is almost entirely depleted in the EMAC simulations at these altitudes while MIPAS still detects maximum values of up to 5 pptv. Possible reasons for this discrepancy discussed below might be (1) measurement errors, or (2) the impact of uncertainties in the parameters of reactions R1 and R2. Further we will address the effect of reaction R3 which is not considered in the present implementation of EMAC.

1. In the top row of Fig. 11, again the mean profile of $BrONO_2$ measured at 5°S is again shown together with the EMAC model results in panel "day". As in the case of the night-time discrepancy, the values of $BrONO_2$ observed during sunlit conditions cannot be explained by the estimated retrieval errors which are about 1–2 pptv at those altitudes (Fig. 1). However, as can be seen in Fig. 11, the mean measured profile values below about 24 km are negative indicating an unresolved issue in the retrieval at these altitudes. Still the negative values are in the range of the estimated uncertainties. To investigate whether the positive values above, where the discrepancy to the model becomes apparent, might be due to an oscillatory feature caused by the negative values below, we have tested different retrieval options (increasing the regularization strength, performing retrievals of log(vmr) instead of vmr, skipping tangent altitudes below). Still in all tests, the maximum values at around 30 km appeared in the retrieved profiles, indicating them as a robust feature.

2. Sensitivity calculations using the 1d model are also shown in Fig. 11. As a baseline for these simulations we have applied the setting "SAD adapted" so that the calculations during night-time fit to the observations also at altitudes below 27 km. The resulting profile from the 1d model run also shows values which are smaller than the observed $BrONO_2$ mixing ratios by about 3 pptv at around 30 km altitude. Though being about 1 pptv larger than the EMAC simulations at those altitudes, the 1d results are also not compatible with the measurements.

   To test the sensitivity on the production of $BrONO_2$ via the three-body reaction R1, we have used the JPL2019 formulation (Burkholder et al., 2019) instead of that by Atkinson et al. (2007) which was applied in the EMAC and 1d baseline runs (Tab. D1). This led to an increase of $BrONO_2$ vmr values by about 1 pptv, see dash-dotted green curve in Fig. 11. Further increasing these rate coefficients by a factor of two, which is well covered by the 2-$\sigma$ uncertainty factor of 3.7 for that reaction at 220 K (Burkholder et al., 2019), lead to an additional increase by about 3 pptv (dash-dotted olive curves in Fig. 11). While these results coincide now with the observed daytime abundances of $BrONO_2$ at around 30 km, the increase at lower altitudes does not correspond well with the observations there. Moreover, around 30 km during night the calculations overestimate the measurements by up to 2 pptv.

   To test the sensitivity with respect to the photolysis of $BrONO_2$ (Eq. R2), we have divided the photolysis rate by 1.2, the 2-$\sigma$ uncertainty as provided by Burkholder et al. (2019). The result is illustrated by the blue dashed curves in Fig. 11. The resulting increase of only around 0.5 pptv is too small compared to the observations. Moreover, the increase appears over a larger altitude range (from 22 up to 34 km) compared to the more confined region between about 27 and 33 km where the increased daytime values are observed.





The grey dotted curve in Fig. 11 illustrates the effect on the simulated mixing ratios when the loss reaction R3 is included in the 1d model by using the reaction coefficients from the JPL2019 compilation, see Tab. D1 (Burkholder et al., 2019).

With a reduction of up to $1.5\,\mathrm{pptv}$ this obviously drives the concentrations further away from the observations.

In conclusion, we have found no unequivocal explanation for the high measured daytime mixing ratios of $BrONO_2$ at around $30\,\mathrm{km}$ over the tropics/subtropics: (a) the differences between models and observations are outside the estimated measurement errors, (b) the uncertainties of the cross sections for photolysis (R2) of $BrONO_2$ are by far too small, (c) the error estimates for reaction R1 would allow a sufficient increase of $BrONO_2$ mixing ratios but over a too large vertical extend, and (d) any

inclusion of reaction R3 opens the gap between simulated and observed $BrONO_2$ concentrations even more.



**Figure 11.** Sensitivity 1d model simulations to test the daytime EMAC model underestimation of $BrONO_2$ around 29 km altitude for $BrONO_2$ (top row), BrO (middle row) and HOBr (bottom row). Left: diurnal evolution of the "1d base" run for a period of two days. The white dashed lines indicate the local solar times of the MIPAS observations. The columns "night" and "day" contain the averaged night-time and daytime profiles of MIPAS at 5°S as black solid lines along with error bars indicating the 2-$\sigma$ estimated measurement uncertainty. The mean EMAC results are provided in red. The "MIPAS" and "EMAC" curves are the same as in Fig. 9. The other curves illustrate the results of the 1d model simulations and the columns "diff night" and "diff day" show the differences between the 1d sensitivity runs and the "1d base" run.





## 4 Estimation of total stratospheric $Br_y$

The MIPAS dataset of $BrONO_2$ allows us to determine the total stratospheric content of bromine ($Br_y$) by using the so-called inorganic method. It is a linear correction of the observed bromine species by multiplication with the ratio between total modelled bromine ($Br_y^{mod}$) and the modelled bromine species observed. This procedure has often been applied in case of

daytime observations of BrO (e.g. Dorf et al., 2006a, b). Wetzel et al. (2017) was the first to apply this method to nightime observation of $BrONO_2$:

$$Br_y = BrONO_2^{meas} \times \frac{Br_y^{mod}}{BrONO_2^{mod}}. \tag{1}$$

Using night-time observations of $BrONO_2$ instead of BrO measured during day to derive $Br_y$ should have the advantage that $BrONO_2$ during night makes up a larger fraction of $Br_y$ than BrO does during day, due to the continuous production of

$BrONO_2$ (Eq. R1).

The modelled night-time ratio between $BrONO_2^{mod}$ and $Br_y^{mod}$ has been examined to decide which region (in terms of altitude and latitude) to use for determination of $Br_y$ from the MIPAS dataset (cf. Fig. C1). Values clearly exceeding 90% are simulated in mid-latitudes mainly during spring/summer/autumn and centred at altitudes around 26 km. Thus, for the analysis we have chosen data at 25–26 km altitude and 40–60°latitude from October to March in the southern and from

April to September in the northern hemisphere. Additionally, due to the low seasonal variability of $BrONO_2$ and to capture relatively young air masses, we considered also tropical data at 29–30 km in the analysis. While not exceeding 90%, the ratio $BrONO_2^{mod}/Br_y^{mod}$ is still larger than 85% (see Fig. C1).

For stratospheric measurements, the values of derived total inorganic bromine are generally assigned to their year of entry into the stratosphere (e.g. Engel et al., 2018, Fig. 1-16). Here we have used an update of the age-of-air dataset determined

from MIPAS retrievals of $SF_6$ (Haenel et al., 2015; Stiller et al., 4–8 May 2020). Typical values of age-of-air in case of the mid-latitude observations range between 5 and 6.5 years and in the tropics between 3.5 and 5 years with errors of about 1 year.

Dependent on the date of stratospheric entry, total $Br_y$ (red) estimated from the MIPAS observations through Eq. 1 in comparison to the original MIPAS observation ($BrONO_2^{meas}$, blue) is shown in Fig. 12. The solid lines represent weighted means of all data points, where for the weighting the a-posteriori estimated random error (dotted black curves in Fig. 1) was

applied. The resulting random error as indicated by the thickness of the line is very small (0.04–0.07 pptv) due to the large amount of data points. The major part of the uncertainty in our estimation of total $Br_y$, however, is due to the combined parameter and systematic error (the green line in Fig. 1) which is shown by the red shading in Fig. 12. It should be noted that this error term is a combination of a variety of estimated uncertainties each of which might also partly be random in nature with different temporal correlation lengths (e.g. correlated over each of the two measurement phases). Thus, we have made a

conservative assumption in considering all those as systematic - but possibly also underestimating them by applying quadratic combination of the single error terms.

Our estimates of total $Br_y$ vary from 21.0±1.4 pptv and 21.4±1.4 pptv for the northern and southern mid-latitude regions (years of stratospheric entry: 1997–2006) to a maximum of 22.4±1.7 pptv in the tropical stratosphere (years of stratospheric





entry: 1998–2007) (Table 1). The values of $Br_y$ from the mid-latitudes of both hemispheres coincide clearly within their

uncertainty ranges. Since it is unlikely that the real values of total $Br_y$ vary strongly in the stratosphere, the difference of 1–1.4 pptv between tropical and mid-latitudinal estimates are more probably caused by uncertainties. These may either be caused by errors in the $BrONO_2$ concentrations derived from MIPAS or be due to the calculation of total $Br_y$ from $BrONO_2$ through of Eq. 1 (or a combination of both). The first explanation would require a retrieval error component varying with latitude e.g. due to some temperature dependence while the second one implies model uncertainty. Since the model adjustment of $Br_y$

from $BrONO_2$ is much larger in the tropical stratosphere (about 2.5 pptv) than at mid-latitudes (about 0.5 pptv), the second explanation would affect more strongly our estimation of $Br_y$ in the tropics.

   $Br_y$ obtained from MIPAS can be compared to data of $Br_y$ derived from observations of BrO, as summarized for example in Fig. 1-16 of Engel et al. (2018). To provide an easy way of comparison, we have replicated the single values of those datasets in each panel of our Fig. 12 and have collected their respective mean values in Table 1, limited to the period of stratospheric

entry from the MIPAS dataset. Obviously, all of these observations are compatible with the values derived from MIPAS and lie clearly within the uncertainty estimates of MIPAS data. The balloon-borne observations ranging from about 20.4 pptv to 21.3 pptv are more in-line with the mid-latitude values of MIPAS, as are the ground-based observations from Harestua with 21.0 pptv. The $Br_y$ value of 22.4 pptv from the ground-based observations in Lauder fits, however, more to the higher tropical MIPAS estimates. Further, the $Br_y$ estimates from $BrONO_2$ measurements during two balloon-flights of the MIPAS-B

instrument (21.6 pptv and 22.7 pptv, respectively) agree with both mid-latitude and tropical MIPAS-Envisat values.



**Figure 12.** Series of averaged MIPAS $BrONO_2$ measurements (dark blue dots) and derived total stratospheric $Br_y$ (red dots) for different altitude and latitude bands over the time of stratospheric entry. Dark blue and red lines indicate the related time averaged mean values over the whole period and the red shading indicates the estimated 1-$\sigma$ uncertainty. The other data points and error bars are estimates of $Br_y$ from observations of BrO taken from Fig. 1-16 in Engel et al. (2018) as well as updates of the MIPAS-Balloon observations of $BrONO_2$ (Wetzel et al., 2017). See also Tab. 1.



**Table 1.** Mean values, standard deviation, the standard error of the mean and the estimated accuracy of total stratospheric $Br_y$ as derived from MIPAS, from Fig. 1-16 of Engel et al. (2018) as well as single observations by the MIPAS-Balloon experiment (Wetzel et al., 2017) during the stratospheric entry years 1997–2007, see also Fig. 12.

| Instrument | Year of stratospheric entry | $Br_y$ derived from | $Br_y$ mean (pptv) | $Br_y$ Std (pptv) | $Br_y$ Std(mean) (pptv) | $Br_y$ Est. accuracy (pptv) |
|---|---|---|---|---|---|---|
| MIPAS[1] (40–60°S) | 1997–2006 | $BrONO_2$ | 21.37 | 3.0 | 0.07 | 1.4 |
| MIPAS[1] (20°S–20°N) | 1998–2007 | $BrONO_2$ | 22.37 | 3.62 | 0.04 | 1.7 |
| MIPAS[1] (40–60°N) | 1997–2006 | $BrONO_2$ | 20.98 | 3.69 | 0.07 | 1.4 |
| Balloon[2,3] (occultation) | 1998–2000 | BrO | 21.25 | 0.0 | 0.0 | 2.5 |
| Balloon[2,3] (occultation, diff. instr.) | Nov 2006 | BrO | 20.5 | | | 3.5 |
| Balloon[2] (Langley) | 1998–2004 | BrO | 20.36 | 0.65 | 0.13 | 2.5 |
| Groundbased[2,4] (Harestua) | 1997–2007 | BrO | 20.97 | 0.48 | 0.04 | 3.8 |
| Groundbased[2,4] (Lauder) | 1997–2001 | BrO | 22.35 | 0.36 | 0.07 | 4.0 |
| MIPAS-Balloon[5] (Kiruna) | Apr 2005 | $BrONO_2$ | 21.6 | | | 2.2 |
| MIPAS-Balloon[5] (Timmins) | Sep 2009 | $BrONO_2$ | 22.7 | | | 1.9 |

[1] This work, [2] WMO (2018), [3] Dorf et al. (2006b), [4] Hendrick et al. (2008), [5] update of Wetzel et al. (2017).





## 5 Conclusions

We have presented the first global dataset of $BrONO_2$ volume mixing ratio profiles for day and night derived from $10°$-zonally and 3-daily averaged MIPAS spectra covering the whole period of observations from 2002–2012. A comparison with EMAC model simulations confirms overall our current understanding of the chemical processes influencing the global zonal mean distribution, as well as the diurnal and seasonal variations of $BrONO_2$ in the stratosphere. Still, remaining differences indicate uncertainties in modelled processes as well as in boundary conditions.

One deviation, the underestimation of $BrONO_2$ concentrations by the model at high latitudes during winter, could be explained. It is caused by the missing additional $NO_x$ source in the model located in the mesosphere and lower thermosphere. The energetic-particle produced $NO_x$ is transported downwards by polar winter subsidence thereby contributing to the production of $BrONO_2$ - a process which we could observe here for the first time. In future, modelling efforts are envisaged to study this effect on the high-latitude bromine budget as well as its impact on stratospheric ozone.

Two further inconsistencies between model and measurement are more difficult to unravel and final explanations remain open. First, a globally present disparity are the higher simulated values in the lower stratosphere especially at night. Sensitivity calculations with our 1d model indicate as the only possible means to decrease $BrONO_2$ concentrations a more efficient heterogeneous loss of $BrONO_2$ e.g. via reaction R4. However, to reach values compatible with the observations, an increase of aerosol surface area densities and/or reaction probabilities would be required. Even for reaction probabilities of unity, aerosol SADs would have to be increased by factors of 2–3 to reproduce the observations of $BrONO_2$. Such an increase in lower stratospheric aerosol SAD would, however, not agree with current satellite and in-situ observations. Another possible cause, a too efficient conversion of organic to inorganic bromine species in the model would be in disagreement with previous balloon and satellite observation of BrO.

Further, the model showed an underestimation of $BrONO_2$ abundances at low latitudes and altitudes of around $27–32\,km$ during daytime. Here, only an increase in the production of $BrONO_2$ (Eq. R1) within its uncertainty range lead to sufficient agreement with the observations at altitudes around $30\,km$, albeit aggravating it below about $27\,km$. Inclusion of Eq. R3, the depletion of $BrONO_2$ via reaction with $O(^3P)$ only increases the difference between model and simulation. It should be noted here that independent information of the reaction parameters for R3 is missing (Burkholder et al., 2019) which might rise concern about its validity.

While we cannot rule out for sure that unaccounted systematic errors in the observations are responsible for these discrepancies, this seems rather unlikely given their overall fit to the model as well as the error assessment. This view is supported by the estimation of the total stratospheric bromine content from MIPAS $BrONO_2$ measurements for years of stratospheric entry between 1997 and 2007, i.e. around the maximum of stratospheric total bromine content (Engel et al., 2018). At mid-latitudes, where the model correction to estimate $Br_y$ from observed $BrONO_2$ volume mixing ratios is smallest, we derived an average value of $21.2\pm1.4\,pptv$ of total stratospheric $Br_y$ which fits very well to independent estimates based on observations of BrO and to $Br_y$ estimates derived from $BrONO_2$ observations of the MIPAS balloon experiment.





In case the inconsistencies between model and observations as discussed above are due to model uncertainties, they could
also affect estimated ozone loss processes through bromine cycles. In future, our dataset of $BrONO_2$ from MIPAS can be
combined e.g. with the simultaneous daytime $BrO$ observations from the SCIAMACHY instrument on Envisat to investigate
the revealed issues about possible deficiencies in our understanding of stratospheric bromine chemistry as well as to gain more
insight in possible uncertainties in the observations.

*Data availability.*  MIPAS level-1b data are provided by ESA (https://earth.esa.int/web/sppa/mission-performance/esa-missions/envisat/mipas/
products-availability/level-1/level1-8.03). SAGE II data were obtained from the NASA Langley Research Center Atmospheric Science
Data Center https://asdc.larc.nasa.gov. In-situ aerosol data were retrieved from http://www-das.uwyo.edu/~deshler/Data/Aer_Meas_Wy_
read_me.htm. The MIPAS $BrONO_2$ dataset and model results are available upon request from the author and at the KITopen repository,
https://doi.org/10.5445/IR/1000136324 (Höpfner et al., 2021).





**Appendix A: Application of averaging kernels on EMAC model results**

The EMAC model data without and with application of the respective averaging kernels from the $BrONO_2$ retrieval are shown in Figs. A1 and A2.



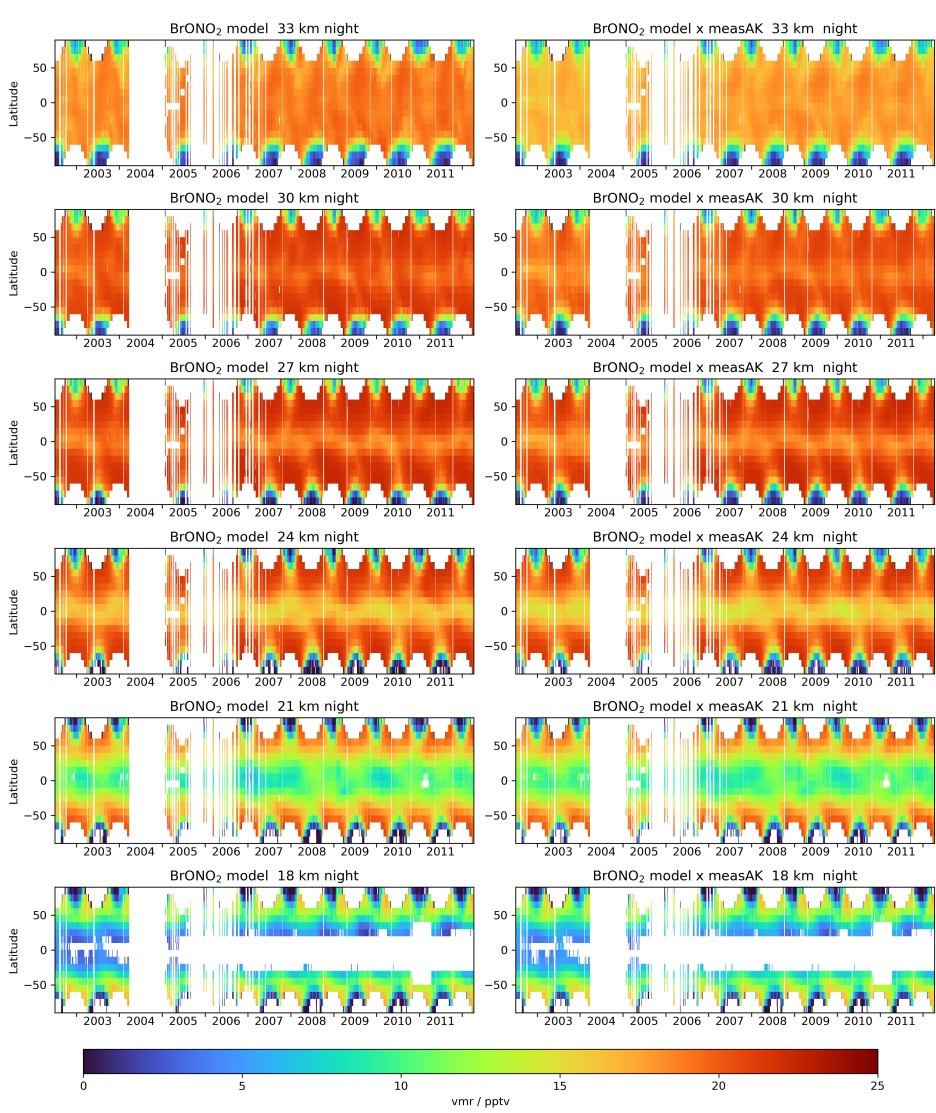

**Figure A1.** Same as in Fig. 3 but showing the pure model results of $BrONO_2$ volume mixing ratios in dark conditions on the left side in comparison to the model results with the retrieval averaging kernels applied on the right side.



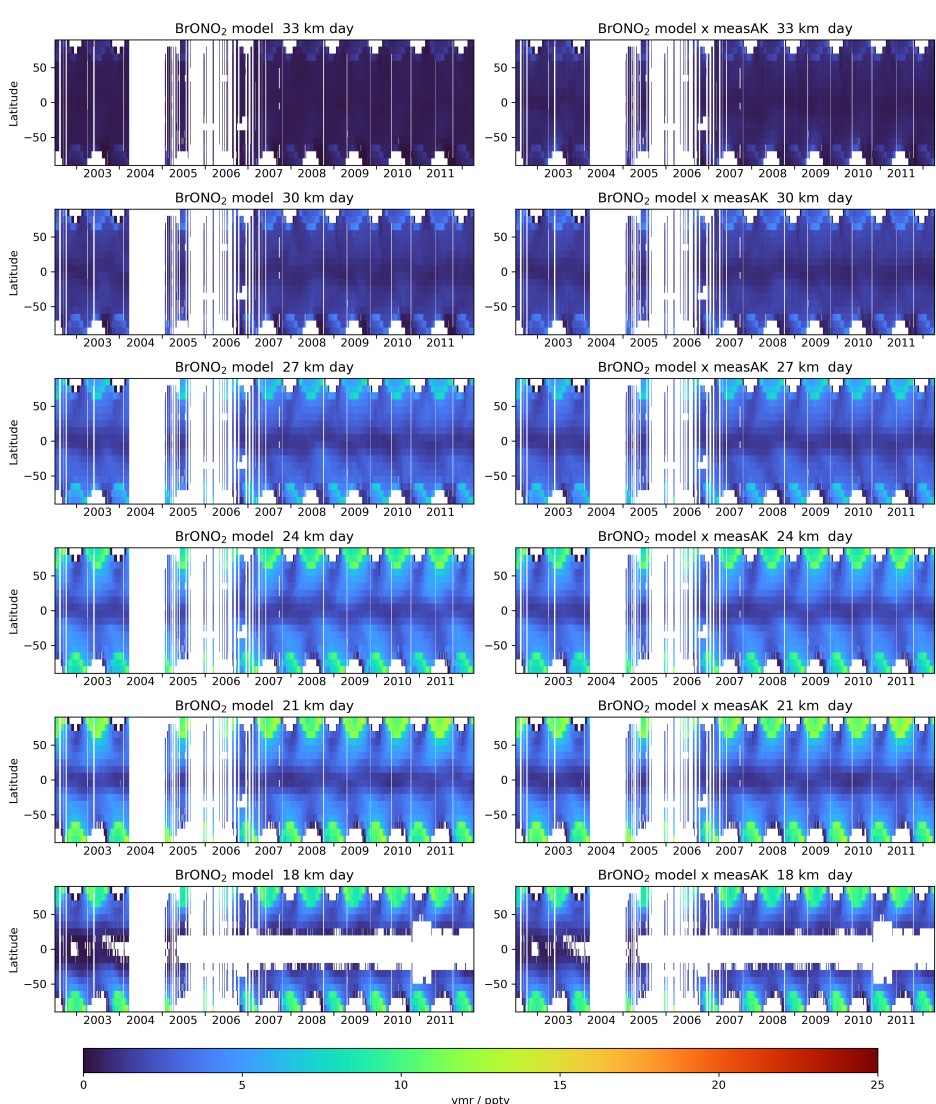

**Figure A2.** Same as in Fig. A1 but for sunlit conditions.



## Appendix B: $NO_2$

Figures B1 and B2 show the results of $NO_2$ volume mixing ratio profiles simultaneously retrieved with $BrONO_2$ in comparison with EMAC model data.



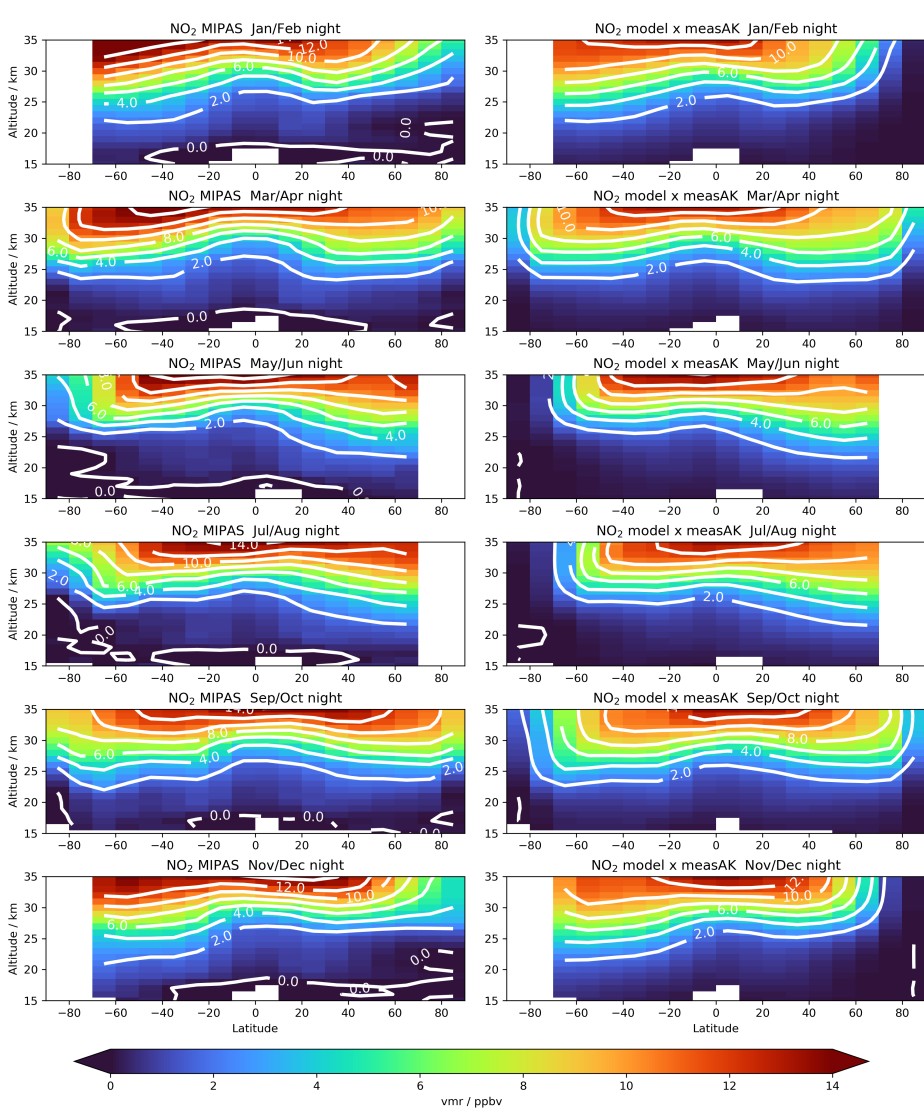

**Figure B1.** Bimonthly averaged cross-sections (altitude versus latitude) of NO$_2$ volume mixing ratios for night-time conditions. Left: measurements, right: model with retrieval averaging kernels applied.



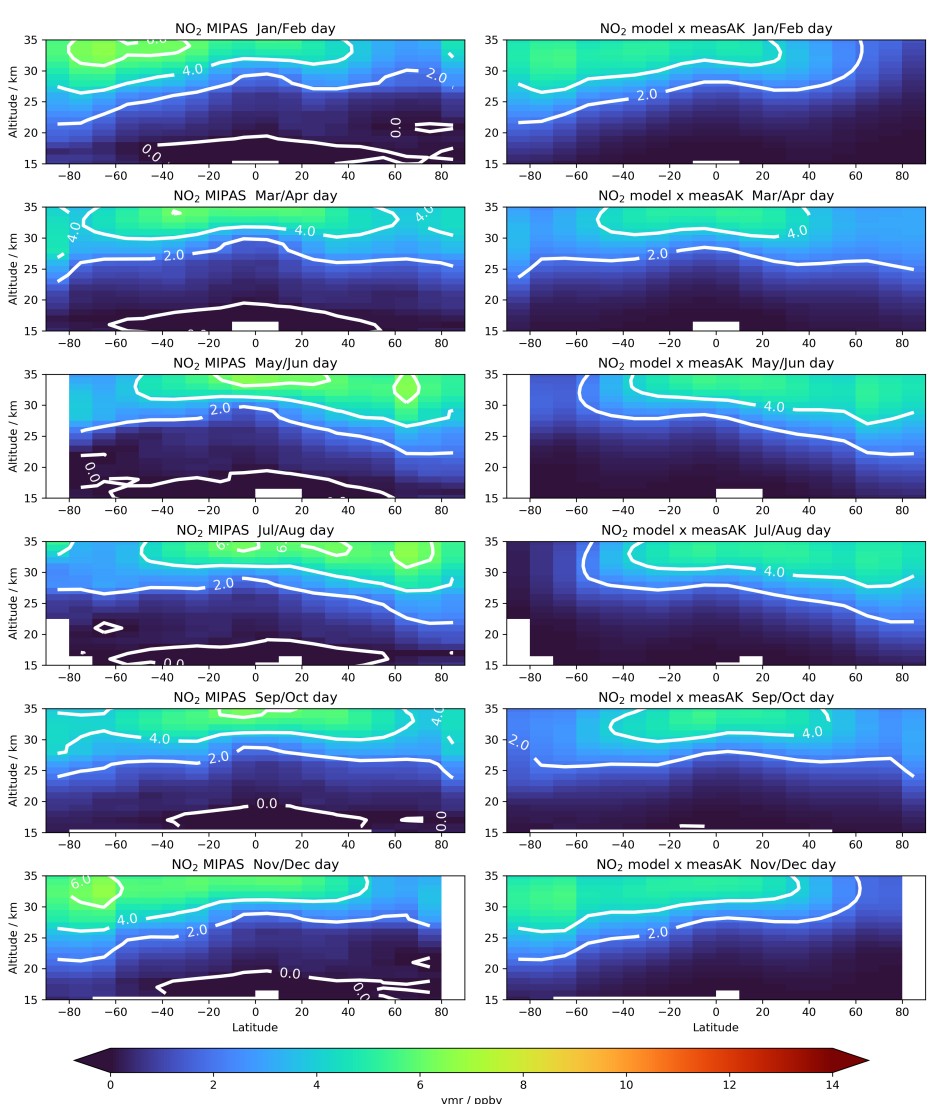

**Figure B2.** Same as Fig. B1 but for sunlit conditions.



## Appendix C: $BrONO_2/Br_y$ ratio

In Fig. C1 the ratio between $BrONO_2$ and total inorganic $Br_y$ from the EMAC model data is shown.





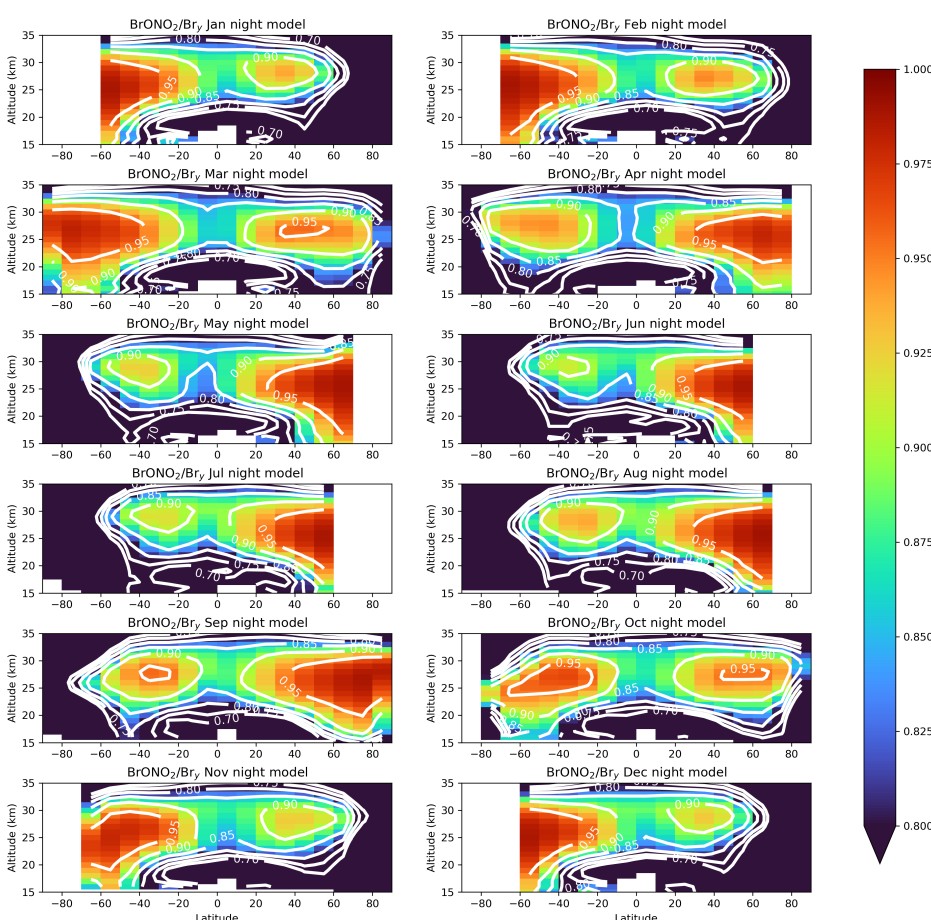

**Figure C1.** Modelled ratio of $BrONO_2$ to $Br_y$ volume mixing ratios averaged over the whole measurement period per month.





## Appendix D: Reaction parameters

**Table D1.** Major bromine reaction parameters as used in the EMAC and the 1d baseline model runs compared to the JPL2019 compilation.

| Reaction | EMAC | JPL2019[1] | 1d base |
|---|---|---|---|
| $BrO + NO_2 \xrightarrow{M} BrONO_2$ | k_3rd_iupac( $k_0$,n,$k_\infty$,m,$f_c$)[2,3] | k_3rd( $k_0$,n,$k_\infty$,m,$f_c$)[4] | EMAC |
|  | 4.7e-31, 3.1, 1.8e-11, 0.0, 0.4 | 5.5e-31, 3.1, 6.6e-12, 2.9, 0.6 |  |
| $BrONO_2 + h\nu \rightarrow$ Products | Burkholder et al. (1995) | Burkholder et al. (1995) | EMAC |
| $BrONO_2 + O(^3P) \rightarrow BrO + NO_3$ | - | 1.9e-11, -215[5] | - |
| $BrONO_2 + H_2O(s,l,H_2SO_4 \cdot nH_2O)$ $\rightarrow HOBr + HNO_3$ | Hanson et al. (1996) | Hanson (2003) | JPL2019 |
| $Br + O_3 \rightarrow BrO + O_2$ | 1.7e-11, 800 [2] | 1.6e-11, 780 | EMAC |
| $BrO + NO \rightarrow Br + NO_2$ | 8.7e-12, -260 [2] | 8.8e-12, -260 | EMAC |
| $BrO + HO_2 \rightarrow HOBr + O_2$ | 4.5e-12, -500 [2] | 4.5e-12, -460 | EMAC |
| $HOBr + h\nu \rightarrow$ Products | Ingham et al. (1998) | Ingham et al. (1998) | EMAC |

[1] Burkholder et al. (2019), [2] Atkinson et al. (2007), [3] $k_0(T) = k_0(300K/T)^n$, $k_\infty(T) = k_\infty(300K/T)^m$, $k_{ratio} = [M]k_0(T)/k_\infty(T)$,
$N = 0.75 - 1.27\log_{10}(f_c)$, $k\_3rd\_iupac = \frac{[M]k_0(T)}{1+k_{ratio}} f_c^{\left(1+\left(\log_{10}(k_{ratio})/N\right)^2\right)^{-1}}$, [4] $k_0(T) = k_0(298K/T)^n$, $k_\infty(T) = k_\infty(298K/T)^m$,
$k_{ratio} = [M]k_0(T)/k_\infty(T)$, $k\_3rd = \frac{[M]k_0(T)}{1+k_{ratio}} f_c^{\left(1+\left(\log_{10}(k_{ratio})\right)^2\right)^{-1}}$, [5] A, E/R, $k(T) = A\exp(-E/RT)$



*Author contributions.* MH performed the MIPAS data retrieval with input from GW, GS and TvC. JO advised on spectroscopy. OK, RR, BMS, GW, SJ and MH performed and supported simulations with EMAC and the 1d model. FH and GS provided age-of-air data from
MIPAS. All the authors contributed to the scientific discussion. MH prepared the manuscript with support from all co-authors.

*Competing interests.* GS and TvC are associate editors of ACP but have not been involved in the evaluation of this paper.

*Acknowledgements.* Provision of MIPAS level-1b calibrated spectra by ESA and meteorological analysis data by ECMWF is acknowledged. SAGE II satellite and in-situ balloon data on aerosol surface area density were obtained from the NASA Langley Research Center Atmospheric Science Data Center and the University of Wyoming, Department of Atmospheric Science (Terry Deshler), respectively. We would
like to thank Klaus Pfeilsticker (University of Heidelberg) and Francois Hendrick (Belgian Institute for Space Aeronomy) for providing data on $Br_y$ from Fig. 1-16 in Engel et al. (2018). The EMAC simulations were performed on the supercomputer ForHLR funded by the Ministry of Science, Research and the Arts Baden-Württemberg and by the German Federal Ministry of Education and Research. We acknowledge support by the German Federal Ministry of Education and Research through the project "Surface Climate Impacts of Halogen Induced Stratospheric Ozone Changes (SCI-HI)", grant 01LG1908A, as part of the programme ROMIC-II (Role of the Middle Atmosphere
in Climate).





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
