# Peer review of "The MIPAS global climatology of $BrONO_2$ 2002–2012: a test for stratospheric bromine chemistry"

_Atmospheric Chemistry and Physics, 2021_

## Author Comment (AC1)

We would like to thank the referee for thoroughly working through the manuscript and for all helpful comments and corrections.
Find below our detailed answers to each of the comments/corrections.

*My only broad comment relates to the discussion of uncertainty in the MIPAS BrONO2 observations as it relates to model discrepancies. This is just a suggestion, but presumably the approach used to quantify uncertainty in MIPAS BrONO2 measurements has been similarly applied to other MIPAS products, for which a larger set of validation/correlative observations exist. For example, if the MIPAS ozone, N2O, CH4, etc. observations are shown to agree with other observations to a degree consistent with the estimated uncertainty in the MIPAS products, does that not speak to the robustness of the uncertainty budget approach? I recognize that, to coin the authors' phrasing, such an argument does not constitute an "unambiguous" assessment, but it could be an additional point to bolster the authors' arguments that the discrepancies are unlikely to be dominated by measurement uncertainties.*

Thanks for this comment. In principle we agree with this statement since we use similar assumptions on error contributions as for the trace-gas retrievals of species with much stronger spectral signals than BrONO2. For these species various studies have shown that the estimated errors are reasonable.
However, one has to keep in mind that due to the very small feature of BrONO2 in the spectrum, we use averaged spectra for the retrieval and systematic error terms not considered might have more impact in that case. Thus, we tend to keep the more cautious formulation of the manuscript.

*Specific comments. Most of these are simply wording suggestions.*

*Line 5: "... into sunlight \*observations\* and observations in the dark..."*

Done

*Line 15: insert "upper stratospheric" before "BrONO2"?*

Since the enhancement compared to the model can be seen to altitudes even below 30 km, we would like to refrain from using the term 'upper stratospheric'

*Line 18: "...the lower stratosphere, a simulated production of BrONO2 that is too low during day as well as strongly..."*

Done.

*Line 19: "additionally" -> "in addition"*

Done.

*Line 24: wonder if "smallest" might be better than "lowest" here.*

Agreed.

*Line 34 and others: The reactions have the molecules in italics, but Equation 1 (page 24) has them in upright font. It would be nice to be consistent (probably*

*more of an issue for the copy editor/typesetter). Similarly, would be good to be consistent on whether the y in Bry is in italics or not.*

Spotted well! We have corrected it.

*Line 36: Not being a kineticist, I was unclear whether some of the numbers quoted on this page for reaction rate uncertainties should have units. From the discussion later in the paper, it is clear that these numbers are indeed "factors". Might it be good to say something like "(i.e. 20% uncertainty)" or something like that for this first one, to make that clearer. Again, feel free to ignore this if what you have is the widely accepted practice.*

We have added the information at the suggested place.

*Line 49: I think "Finally" might be better than "At last"*
Done.
*Line 64: "contributing" -> "contribute"*
Done.
*Line 66: comma needed after "changes" I think.*
Done.
*Line 68: Put commas before and after "in consequence"*
Done.
*Line 81: "years" -> "year"*
Done.
*Line 95: "retrieval" -> "retrievals". Also, "has" -> "have"*
Done.
*Line 99: "shortly" -> "briefly"*
Done.
*Line 103: comma needed after averaging.*
Done.
*Figure 1 caption: I suggest "...from four 3-day periods, two during P1 (top) and two during P2 (bottom), for both dark (left) and sunlit (right) conditions"*
Done.
*Line 182: "following" -> "subsequent years". Also, "until" -> "through"*
Done.
*Line 183-184: I suggest: "The coverage at lower altitudes is determined by the lower limit of 15 km, chosen to continue the retrievals to the stratosphere, and by the presence of high-altitude clouds and the scan pattern of MIPAS (which is mainly a factor in the tropics)." Or something like that (move the parenthetical point at the end earlier if it only applies to the clouds).*
Done.
*Line 189: Insert "a" between "as" and "supply"*
Done.
*Line 190: "at PSC particles" -> "on PSC surfaces"? (unless the reactions take place inside the PSC particles, in which case "in PSCs" might be simpler)*
Done.
*Line 237: insert "of BrONO2" before "might". Also ".might be the emission scenario* used for organic bromine species, taken from Warwick. "
Done.
*Line 242: "at" -> "on"? (see above though)*
Done.
*Line 266/267: move "quite well" to the end of the sentence.*

Done.

*Line 273: Perhaps change "those altitudes" to "the 30 km region" to be that bit clearer. Also, comma needed after "errors"*

Done.

*Line 274: Comma needed after "negative"*

Done.

*Line 278/279: Suggest you move "still" from the start of the sentence to before "appeared"*

Done.

*Line 280: Suggest comma after "simulations"*

Done.

*Line 285: Suggest: "The test the sensitivity of modeled BrONO2 to the production..."Line 286: "which" -> "that"*

Done.

*Line 304: "extend" -> "extent"*

Done.

*Line 368: "are" -> "is"*

Done.

*Line 373/374: Suggest "a too efficient" -> "an overly efficient".Line 374: Also, comma needed after "model"*

Done.

*Line 399: Suggest "In case" be changed to "If it is the case that". Also "they" -> "these inconsistencies".*

Done.

*Line 481: Funny spacing in NOy*

Done.

*Line 561: Funny spacing in Bry*

Done.

*(Note, I didn't to a comprehensive check of the references, it's just that those two caught my eye).*

---

## Author Comment (AC2)

We would like to thank the referee Rafael Pedro Fernandez for his in-depth comments and suggestion which helped a lot to improve the manuscript. Find below our detailed answers to each of the comments/corrections.

- *The total stratospheric Bry value of 21.2 ± 1.4 pptv of Bry at mid-latitudes has been estimated for the 40-60S and 40-60N but without considering the 10° latitudinal bins used in previous sections. Wouldn't it help to evaluate if there is any Stratospheric Bry trend and/or sink by means of computing Bry at different latitudes and heights? In particular, Section 4 (Fig. 12) would benefit of a discussion comparing the almost negligible trend in MIPAS Bry trend between 1996-2007 with the reported value in WMO of -0.16 ± 0.07 ppt for the 2004-2014 period.*

  Thanks for this suggestion. We agree that it could make sense to investigate these latitude/altitude/time dependent distributions in much more detail in future work. For the present manuscript we have mainly concentrated on showing examples and their variability regarding the derived total stratospheric Bry from three different latitude/altitude regions (see Fig. 12) to judge how/if these estimations are in general agreement with Bry derived from BrO observations. This can also be regarded as kind of a first, still quite indirect validation of our retrievals. Still, as suggested by the referee, we will provide the more detailed view on the time series for the 10° latitude bins as a supplement to the manuscript and have included it also here at the end (Figures 1, below). Furthermore, deriving trends is also beyond the scope of this work as we see it. Because the time span of the MIPAS observations (with respect to the derived year of stratospheric entry) is just around the maximum of the stratospheric bromine content, a simple trend may even not be the suitable description for the time series.

- *The authors suggest that the low modeled polar BrONO2 during winter is caused by a low bias in NO2 abundance due to missing mesospheric NO2 production in the model (dif_1). However, for dif_2 and dif_3, they highlight different competing processes, but they do not mention even once that an erroneous modeled NO2 abundance could also be affecting the model-observation disagreement. I think this should be explicitly mentioned for all cases. In particular, for dif_2, Barrera et al., (2020) highlighted that VSL bromine impacts in the mid-latitudes depend on the recycling efficiency of ClONO2 and HCl in the lowermost stratosphere. Could modeled ClONO2 recycling also be affecting NO2 abundance in the lower stratosphere and indirectly affecting the modeled BrONO2 sinks?*

  As can be seen in Figs. B1 and B2 of the manuscript's appendix, there is quite a broad consistency between the retrieved and modelled mixing ratios of NO2. The main differences are the nighttime values at high winter latitudes, as proposed in our manuscript as reason for the model's underestimation of BrONO2 there. We have not mentioned the aspect of wrongly modelled NO2 mixing ratios as possible explanation in relation to dif_2 and dif_3 since NO2 agrees quite well for these cases. To show this clearer, we have added plots equivalent to Figures 9 and 11 of the manuscript but for NO2 (see Figure 2 below): these show a very good agreement of modelled and measured NO2 up to 26 km altitude. Thus, dif_2 is very probably not due to wrong modelling of NO2. Above 26 km altitude, there are some deviations in NO2 mixing ratios,

however, at most 20% between the measurements and EMAC or the 1d-model. These deviations cannot account for the daytime model underestimation of BrONO2 (dif_3) – the slight overestimation of NO2 by EMAC around 30 km would even point toward the wrong direction. We have added a paragraph in the manuscript describing the role of NO2 for dif_2 and dif_3 accordingly and add the related Figures to the NO2 overview Figures in Appendix B.

*Minor comments:*

*P4, L109: "the related a-priori profile for the target species BrONO2 was set to zero": I'm not an expert in this field, but I thought that for satellite retrievals it was necessary to include a non-zero a-priori profile.*

In case of first order Tikhonov-regularisation using volume mixing ratios (and not log(vmr)) as retrieved quantities, a zero a-priori profile is generally possible. However, in our case the referee is right that we have actually used an altitude-constant a-priori profile with BrONO2 mixing ratios of 0.1 pptv. We have corrected this in the manuscript.

*P5, L138: "Around that, the blue shading indicates the variability of the estimated errors between all latitude bands". Please rephrase. Does "that" points at the mean total error? Is it "between" of "for" all latitudes?*

Sorry that the sentence was not clear. It is corrected to 'Around this total error estimate, the blue shading indicates the variability of the estimated errors for all latitude bands'.

*Fig. 1: Rapid eye-reading the figure, it is evident that below 20 km, Tot parameter errors and spectral noise are the dominant contribution to the total error, while at higher heights it is mostly dominated by spectral noise. A simple sentence on the text highlighting this would be useful.*

We have added this information in the manuscript.

*P8, L167: "During the MIPAS measurement periods, from the model output first all data within one hour around 10 LT and 22 LT were selected. Depending on their latitude, longitude and altitude, they were then assigned to day and night-time conditions and averaged over the observational bins of 10 latitude and three-day periods". Do you mean that model output was filtered for specific hours to match MIPAS observations? And also, did you consider any additional condition to filter day/night time values very close to twilight conditions where the radiation intensity is reduced (mostly at high latitudes).*
*Please make it clear.*

Due to the sun-synchronous orbit of Envisat, MIPAS measurements took place around 10 am/pm local solar time. Thus, the selection with respect to local solar time 'follows' the trace of the satellite. Then the assignment to sunlit or observations in the dark is made through calculating the solar elevation at each point. We have made this clearer in the revised manuscript. The referee is also

right that at high latitudes, scenes with solar elevation angles near zero might be included and be difficult to interpret. However, the considerations and conclusions made in the manuscript refer to situations which are not affected by such solar geometries. We have added in the text a note of caution about such scenes.

*P9, L193: Do you mean a "seasonal" signal instead of annual? What do you mean by outstanding?. Finally, it would be useful to provide a couple of sentences summarizing theMaximum and Minimum values observed for different heights and latitudes before gettinginto the MIPAS-Modelling comparison.*

Sorry for the unclear formulation: We have changed it to 'The lack of a similarly clear seasonal variability at tropical latitudes.' We have added information about the observed mixing ratios as recommended by the referee.

*P16,L198: What is the Averaging Kernel Matrix? A kernel including the 10 LT and 22 LThours? If that is the case, please make it consistent to the description in Section 2.2.*

Sorry for the misunderstanding. What we mean here are the vertical averaging kernel matrices which are diagnostics characterizing the MIPAS profile retrievals of BrONO2 regarding vertical resolution. We refer to the averaging kernels in section 2.1 with the citation of the book by Rodgers (2004). We have tried to make this clearer in the new version of the manuscript by changing the sentence to: 'To take into the account the limited vertical resolution of the measurements for these comparisons, we have applied the averaging kernel matrix of each retrieved profile of BrONO2 (see Sect. 2.1) to the related modelled profile'.

*P16, L220: "Such differences are also present during day, albeit to a smaller extent (up toabout 4 pptv, see Fig. 8)." … due to the smaller BrONO2 abundance. Clearly, as BrONO2 levels are higher during the night, then the absolute differences will be higher during the night than during the day, where BrONO2 vmr are smaller. I wonder if the relative/percentage difference between model and observations are similar during the day  than during the night? Independently of the answer, this could be explicitly mentioned in the text.*

Yes, during day the relative differences between measurement and model at around 20 km in mid-latitudes and 22-25 km in the tropics are similar to nighttime relative differences with about 50%. Below these altitudes, the daytime relative differences become much larger due to the very small values of BrONO2 there and are difficult to compare. We have added a related sentence in the revised manuscript.

*Fig. 9: The "diff night" and "diff day" panels show a very similar profile, though the absolute variation is considerably different. Have you estimated if the relative/percentage difference between day/night are similar?*

We have replotted Fig. 9 with relative differences (see Figure 3 below): the nighttime effect of changing SAD on BrONO2 mixing ratios is about a factor of 10 larger in relative terms compared to the daytime effect.

*P24, Eq.1: You have only applied the expression to night-time modelled values. Have you performed the same analysis with daytime values? If so, did you find any difference worth to be mentioned?*

We have not performed the analysis on daytime measurements. Our goal was to get the best possible estimate of Bry which we think can best be estimated when the correction (Eq.1) is smallest. For a much more in-depth analysis it could be a good idea to inspect also BrO values derived from daytime measurements (e.g. in combination with the Sciamachy dataset of BrO).

*P25, L345-346: "Since the model adjustment of Bry from BrONO2 is much larger in the tropical stratosphere (about 2.5 pptv) than at mid-latitudes (about 0.5 pptv), the second explanation would affect more strongly our estimation of Bry in the tropics." What do you mean here? That the estimated Bry will be more realistic for high BrONO2/Bry ratios?*
*Wouldn't it be worth to compare the Bry estimation for each latidunal bin of 10° at different seasons, and to determine if the estimated Bry stratospheric abundance for each band is consistent?*

The 'second' explanation refers to the application of Eq. 1, i.e. the correction using the modelled Bry/BrONO2 ratio. We will make the sentence clearer in the revised manuscript. For a discussion on Bry estimation from each latitude bin: see our comment above under the 1st major issue.

*Fig. 12: I found a bit confusing that the colored symbols with measurements (other than MIPAS) at different locations are shown in all 3 panels. Wouldn't it be better to include in each panel only those observation corresponding to the latitudinal band where it was measured?*

We have repeated the non-MIPAS measurements in all panels to guide the eye. Therefore, we would like to leave them in all panels. However, in the new plot as shown in Figure 4 below, now we highlight the location (south, tropics, north) of the related observations by larger symbols.

*Fig. B1 and B2: It could be useful to show the modelled NO2 difference in a 3rd column on the right, as it has been shown for BrONO2 in the main text.*

Has been done: see Figures 5 and 6 below.

*Would it be worth to include the estimated AOA in the published dataset?*

We will add the information that this dataset is available on request from Gabriele Stiller (gabriele.stiller@kit.edu).

*Language editing comments:*

*P1,L10-11: Rephrase item (1) in the abstract.*
It is not clear to us why (1) should be rephrased. Thus, we tend not to change the wording.

*P2, L33: ist produced à is produced*
Corrected.
*P2, L35: coupled strongly à strongly coupled*
Done.
*P3, L69: VSLS should be defined after it first usage in L64.*
Done.
*P4,5: ESA, HITRAN, ECMWF, etc. acronyms are not defined.*
Done.
*P16, L209: "values of less than" à "values smaller than"*
Done.
*P17, L259: Define SAD as it first usage in L244.*
Done.
*P22, L299: Tab. D1 à Table D1*
Done
*P24, L327: Dependent à Depending*
Done.
*P24, L337: "vary from 21.0 +/- 1.4 pptv and 21.4 +/- 1.4 pptv for the
northern andsouthern mid-latitude regions", respectively.*
Done.
*P25, L350: Replace Obviously by Notably or other word … as it is not obvious
thatmeasurements performed with different instrument will provide equivalent
results.*
Done.

*References*

*Barrera, J. A., Fernandez, R. P., Iglesias-Suarez, F., Cuevas, C. A., Lamarque,
J.-F., and Saiz-Lopez, A.: Seasonal impact of biogenic very short-lived
bromocarbons on lowermoststratospheric ozone between 60° N and 60° S
during the 21st century, Atmos. Chem.
Phys., 20, 8083–8102, https://doi.org/10.5194/acp-20-8083-2020, 2020.*

[Figure]

Figures 1 below are equivalent to Fig. 12 of the manuscript but for bins of 10°
latitude and 2 km altitude: series of averaged MIPAS BrONO2 measurements (dark
blue dots) and derived total stratospheric Bry (red dots) for different altitude and
latitude bands over the time of stratospheric entry. Dark blue and red lines indicate
the related time averaged mean values over the whole period and the red shading
indicates the estimated 1-sigma uncertainty.

[Figure]

[Figure]

[Figure]

[Figure]

Figure 2: Top: equivalent to Fig. 9 of the manuscript, bottom: equivalent to Fig. 11 of the manuscript, but showing altitude profiles of NO2 volume mixing ratios.

[Figure]

Figure 3: Equivalent to Fig. 9 of the manuscript, but showing relative differences in the 3$^{rd}$ and 5$^{th}$ panel.

[Figure]

Figure 4: Equivalent to Fig. 12 of the manuscript, but indicating the latitudes (south of 20S: top; 20S-20N: middle; north of 20N: bottom) of the non-MIPAS observations by the larger symbol size.

[Figure]

Figure 5: Equivalent to Fig. B1 of the manuscript, but showing also the absolute differences between modelled and measured NO2 mixing ratios as 3rd column.

[Figure]

Figure 6: Equivalent to Fig. B2 of the manuscript, but showing also the absolute differences between modelled and measured NO2 mixing ratios as 3rd column.

[Figure]